# Enhancing the Biological Functionality of Hydrogels Using Self-Assembling Peptides

**DOI:** 10.3390/biomimetics10070442

**Published:** 2025-07-04

**Authors:** Woo Hyun Kwon, Kyoung Choi, Sang Jun Park, GeumByeol Park, Cho Young Park, Yoo Han Seo, Chun-Ho Kim, Jun Shik Choi

**Affiliations:** 1Laboratory of Tissue Engineering, Korea Institute of Radiological and Medical Sciences, Seoul 01812, Republic of Korea; woohyun@kirams.re.kr (W.H.K.); choik1231@kirams.re.kr (K.C.); sjpark@kirams.re.kr (S.J.P.); goldstar810@kirams.re.kr (G.P.); cypark@kirams.re.kr (C.Y.P.); yhseo@kirams.re.kr (Y.H.S.); 2Department of Materials Science and Engineering, Yonsei University, Seoul 03722, Republic of Korea; 3Program in Biomicro System Technology, Korea University, Seoul 02841, Republic of Korea; 4Department of Biomedical and Chemical Engineering, The Catholic University of Korea, Bucheon-si 14662, Gyeonggi-do, Republic of Korea; 5Department of Bioengineering, Hanyang University, Seoul 04763, Republic of Korea

**Keywords:** hydrogel, self-assembling peptides, anti-inflammatory, antimicrobial, anticancer, bioimaging

## Abstract

Hydrogels are ECM-mimicking three-dimensional (3D) networks that are widely used in biomedical applications; however, conventional natural and synthetic polymer-based hydrogels present limitations such as poor mechanical strength, limited bioactivity, and low reproducibility. Self-assembling peptides (SAPs) offer a promising alternative, as they can form micro- and nanostructured hydrogels through non-covalent interactions and allow precise control over their biofunctionality, mechanical properties, and responsiveness to biological cues. Through rational sequence design, SAPs can be engineered to exhibit tunable mechanical properties, controlled degradation rates, and multifunctionality, and can dynamically regulate assembly and degradation in response to specific stimuli such as pH, ionic strength, enzymatic cleavage, or temperature. Furthermore, SAPs have been successfully incorporated into conventional hydrogels to enhance cell adhesion, promote matrix remodeling, and provide a more physiologically relevant microenvironment. In this review, we summarize recent advances in SAP-based hydrogels, particularly focusing on their novel biofunctional properties such as anti-inflammatory, antimicrobial, and anticancer activities, as well as bioimaging capabilities, and discuss the mechanisms by which SAP hydrogels function in biological systems.

## 1. Introduction

Hydrogels are three-dimensional cross-linked macromolecular networks primarily composed of water. Because their structure closely resembles the extracellular matrix (ECM), hydrogels are widely utilized in tissue engineering research. Currently, a variety of biological materials, including natural and synthetic polymers such as collagen, gelatin, hyaluronic acid, sodium alginate, polyethylene glycol (PEG), and polyvinyl alcohol (PVA), are used in the preparation of hydrogels [1,2,3].

Hydrogels serve as a versatile platform for a broad spectrum of biomedical applications, particularly when their structure and function are rationally engineered through materials engineering to modulate key physicochemical properties—such as stiffness, pore size, viscoelasticity, microarchitecture, degradability, ligand presentation, and stimulus responsiveness—which in turn influence cell signaling pathways and cellular fate. Owing to these adjustable characteristics, hydrogels have been extensively utilized in various biomedical fields, including wound dressings, injectable cell delivery systems, controlled drug release platforms, and three-dimensional scaffolds for tissue regeneration of bone, cartilage, skin, and other tissues [4,5,6,7].

Despite their widespread application in biomedical fields, conventional hydrogels exhibit inherent limitations. Natural polymer-based hydrogels demonstrate good biocompatibility and ECM-like characteristics but suffer from poor mechanical strength and inconsistent batch-to-batch reproducibility. In contrast, synthetic polymer-based hydrogels offer precise control over mechanical properties but lack intrinsic biological functionality. Moreover, it is difficult to control the micro- and nanoscale structural organization of these hydrogels, which limits reproducibility and impairs their ability to guide cellular behavior in physiologically relevant environments [8,9,10,11,12].

To address these challenges, self-assembling peptides (SAPs) have emerged as a promising class of biomaterials capable of forming hydrogels or peptide-modified hydrogels through spontaneous, non-covalent interactions, including hydrogen bonding, hydrophobic interactions, and π–π stacking. In addition, SAPs can be engineered to respond to specific stimuli such as pH, ionic strength, enzymatic cleavage, or temperature, enabling dynamic modulation of their assembly and degradation in response to biological cues. These peptides can be precisely designed at the amino acid sequence level to self-assemble into well-defined nanostructures, such as nanofibers, β-sheet-rich networks, micelles, vesicles, or nanotubes, thereby closely mimicking the architecture and biochemical signals of the native ECM (Figure 1) [13,14,15,16].

Through rational sequence design, SAPs can be precisely engineered to exhibit tunable mechanical properties, controlled degradation kinetics, and multifunctionality. In the context of peptide-based biomaterials, multifunctionality refers to the capacity of a single self-assembling peptide (SAP) system to perform multiple biological and physicochemical functions simultaneously or sequentially. This attribute is particularly advantageous for biomedical applications such as tissue engineering, drug delivery, and regenerative medicine, where a single material platform must address multiple therapeutic needs. For instance, self-assembling peptides can be rationally designed to provide structural support through the formation of nanofibers or hydrogels, while simultaneously incorporating functional motifs that promote cell adhesion, modulate inflammation, deliver drugs, or respond to environmental stimuli such as pH, enzymes, redox conditions, and light [17,18,19,20,21]. Multifunctionality is typically achieved through the modular design of peptide sequences. Bioactive domains, such as the RGD motif for cell adhesion or sequences with antimicrobial or anti-inflammatory properties, can be incorporated into the self-assembling core without disrupting supramolecular assembly. In addition, responsive elements—such as MMP-cleavable sites or disulfide bonds—can be integrated to enable site-specific drug release or degradation in pathological microenvironments [22,23]. This design strategy allows SAPs to act not only as passive carriers but also as active therapeutic platforms capable of dynamic interactions with the biological environment [24,25].

This approach allows for the direct incorporation of various bioactive motifs—such as the RGD peptide (a cell adhesion motif recognized by integrin receptors), IKVAV (a laminin-derived sequence promoting neurite outgrowth and neural differentiation), and MMP-sensitive sites (matrix metalloproteinase-cleavable sequences that permit cell-mediated remodeling)—without the need for complex chemical conjugation. In addition to forming stand-alone hydrogels, SAPs have also been successfully incorporated into both natural and synthetic conventional hydrogel matrices to enhance their biological functionality and structural complexity. This hybrid strategy combines the favorable mechanical or rheological properties of bulk hydrogels with the molecular-level bioactivity and self-assembling characteristics of peptides. These composite systems improve cell adhesion, promote matrix remodeling, and create more physiologically relevant microenvironments. This broadens the design possibilities for therapeutic and diagnostic applications, including tissue regeneration, drug delivery, and biosensing [26,27,28,29].

This review aims to explore the emerging biofunctional properties of SAP-based or SAP-modified hydrogels, including their anti-inflammatory effects, antimicrobial activity, anticancer potential, and bioimaging capabilities. In addition, we discuss the underlying physicochemical principles and mechanical characteristics that govern how these hydrogels perform and interact within biological systems.

## 2. Bioactive Properties of Self-Assembling Peptides

### 2.1. Anti-Inflammatory

#### 2.1.1. Mechanism of Anti-Inflammatory Peptides (AIPs)

Inflammation is recognized as a normal part of the host response to injury or infection. The human immune and inflammatory responses are regulated by a complex interplay of various systemic factors. Increased blood flow and enhanced capillary permeability allow immune cells and molecules—such as leukocytes, antibodies, cytokines, and complement proteins—to migrate toward the site of immune activation, injury, or infection. Essentially, the inflammatory process marks the initial phase of the immune response against toxins, invading pathogens, allergens, or damaged tissues.

The inflammatory response is initiated when immunological, microbial, or toxic stimuli activate a range of cellular and humoral components. This activation leads to the release of various cell-derived mediators, including cytokines from the interleukin (IL) family, tumor necrosis factor-alpha (TNF-α), prostaglandins (PGs), nitric oxide (NO), reactive oxygen species (ROS), and leukotrienes (LTs). Accordingly, pharmacological inhibition of IL-family cytokines has emerged as a key therapeutic strategy for treating and managing immune and inflammatory disorders. Certain anti-inflammatory agents can suppress the nuclear factor kappa B (NF-κB) signaling pathway, thereby preventing the expression of a broad array of genes involved in pathological processes (Figure 2) [30,31].

In this context, anti-inflammatory peptides (AIPs) constitute a promising class of biomaterials that leverage supramolecular chemistry and immunomodulatory functions to deliver targeted and sustained therapeutic effects. They modulate inflammation through various mechanisms, such as suppression of receptor-mediated signaling, mitigation of oxidative stress, and stabilization against protease degradation [32,33].

More specifically, AIPs play a crucial role in regulating inflammatory responses. First, they suppress inflammation by blocking several steps in lipopolysaccharide (LPS) recognition, signaling, and receptor activation. This includes neutralizing LPS and inhibiting the interaction between lipopolysaccharide-binding protein (LBP) and LPS. Second, AIPs inhibit major intracellular signaling pathways that drive inflammation. They suppress the activation of the NF-κB pathway—a central inflammatory pathway activated via toll-like receptors (TLRs)—leading to the downregulation of cytokines such as TNF-α, IL-1β, and IL-6. Additionally, they inhibit phosphorylation within mitogen-activated protein kinase (MAPK) pathways, including JNK, p38, and ERK, thus reducing the production of inflammatory mediators. They also prevent the internalization of TLR4, further dampening inflammatory signaling. Finally, AIPs modulate the recruitment and phagocytosis of inflammatory cells, and phagocytosis of inflammatory cells, stabilizing the local immune environment and limiting the infiltration of inflammatory cells [34].

Supporting these mechanistic insights, Song et al. reported that the OA-GL12 (GLLSGINAEWPC) peptide significantly promoted wound healing in a full-thickness skin wound model. Inflammatory analysis revealed that OA-GL12 induced the secretion of TNF and transforming growth factor β1 (TGF-β1), resulting in enhanced tissue regeneration. This process involved the activation of the epidermal growth factor receptor (EGFR) [35].

Moreover, macrophages are key immune cells involved in various physiological and pathological processes, including inflammatory disorders. The two major phenotypes are M1 (classically activated) and M2 (alternatively activated) macrophages. M1 macrophages exhibit pro-inflammatory properties and respond to initial injury by secreting cytokines such as TNF-α and IL-1β. In contrast, M2 macrophages are activated by stimuli such as IL-4 and IL-13, as well as secrete anti-inflammatory cytokines like IL-10, thereby suppressing inflammation and promoting tissue regeneration. AIPs can modulate the polarization of macrophages from the M1 to M2 phenotype and may represent a more cost-effective strategy for inflammation control compared to full-length proteins. AIPs are easily conjugated to scaffolds or nanoparticles, are relatively inexpensive to produce, and are more stable than proteins [36].

As another example of immunomodulatory peptides, Deng et al. investigated the effects of vasoactive intestinal peptide (VIP) in a collagen-induced arthritis (CIA) model. VIP significantly reduced the incidence and severity of arthritis and delayed disease onset. It increased the population of regulatory T cells (Tregs) while suppressing pro-inflammatory Th17 and Th1 cells, leading to reduced levels of inflammatory cytokines such as IL-17 and IL-22. Furthermore, VIP inhibited Th17-associated transcription factors (STAT3 and RORγt) and upregulated Foxp3, a key Treg marker [37].

#### 2.1.2. Peptides for Enhancing Anti-Inflammatory Properties in Hydrogel

Recent studies have demonstrated that the incorporation of AIPs into hydrogel systems significantly enhances their immunomodulatory potential, particularly by modulating macrophage behavior and the local cytokine environment. These peptide-based hydrogels often exhibit bioresponsive characteristics, such as pH or enzyme sensitivity, and have shown promise in the treatment of chronic wounds, autoimmune diseases, and inflammation-related tissue degeneration. Below, several representative studies illustrate diverse strategies used to design and apply AIP-functionalized hydrogels across a range of biomedical applications.

Supporting this concept, Kong et al. discussed a study on a self-assembling peptide (SAP) hydrogel (BKF) designed to modulate macrophage polarization within the tumor microenvironment. This nanofibrous hydrogel, formed from a short peptide containing an innate defense regulator motif, multivalently engaged macrophage receptors. The system effectively polarized M0 macrophages into pro-inflammatory M1 macrophages via NF-κB pathway activation and reprogrammed immunosuppressive M2 macrophages into M1-like cells. This strategy shows promise for remodeling the tumor immune microenvironment and enhancing immunotherapy by overcoming immune resistance through macrophage re-education (Figure 3) [38].

Moreover, Sun et al. reported peptide-based composite hydrogels (CRP) for spinal cord injury (SCI) repair. They combined chitosan, RADA16 nanofibers, and a nerve-promoting peptide (PPFLMLLKGSTR) to simulate a microenvironment conducive to neural repair. The hydrogels exhibited excellent injectability, thermo-sensitivity, and biocompatibility, forming stable 3D scaffolds in situ at body temperature. CRP hydrogels promoted the proliferation and differentiation of endogenous neural stem cells, reduced neuroinflammation, and protected neurons. In vivo studies demonstrated improved motor recovery in SCI rats, likely through activation of the PI3K/AKT/mTOR signaling pathway. This strategy offers a minimally invasive and effective platform for SCI treatment, with strong potential in clinical translation and tissue engineering applications [39].

Targeting inflammation in autoimmune disease, Wu et al. developed BiPM@IOK, a pH-sensitive injectable peptide hydrogel for rheumatoid arthritis (RA) therapy. By integrating BiNS/PEI nanoparticles and methotrexate (MTX) into the hydrogel matrix, they enabled collapse under acidic conditions typical of the RA microenvironment. MTX suppressed pro-inflammatory cytokines in macrophages, while BiNS/PEI enabled photothermal and photodynamic therapy to eliminate pathogenic fibroblast-like synoviocytes. The results showed effective suppression of RA symptoms, synovial hyperplasia reduction, and cytokine level control without systemic toxicity (Figure 4a,b) [40].

Regarding gas signaling modulation, Younis et al. discussed the limitations of macromolecular hydrogels, such as low payload capacity and rapid nitric oxide (NO) release, and addressed them by incorporating FFc5FF peptides using a sticker-spacer strategy into a nanofibrous hydrogel. The peptide not only facilitated NO generation but also effectively sequestered exogenous NO gas, thereby suppressing the secretion of pro-inflammatory cytokines, including IL-6 and TNF-α [41].

Meanwhile, Wang et al. introduced a biocompatible and stable hydrogel based on the KLD1R peptide (Ac-KLDLKLDLKLDLR-CONH_2_) to address impaired bone healing in diabetic patients. By loading interleukin-10 (IL-10) into the hydrogel, they overcame limitations associated with their short half-life and low local concentration of cytokines. The sustained release of IL-10 promoted M2 macrophage polarization, attenuated inflammation, and enhanced the osteogenic differentiation of mesenchymal stem cells (MSCs) [42].

Focusing on cytokine inhibition, Liao et al. synthesized the SAP, Nap-DFDFDEGPIRRSDS (termed FP), that effectively inhibited tumor necrosis factor-alpha (TNF-α), a key cytokine in rheumatoid arthritis pathology. This peptide, prepared by solid phase peptide synthesis (SPPS), demonstrated rapid inhibition of TNF-α within 15 min and reduced polarization of pro-inflammatory M1-type macrophages in an adjuvant-induced arthritis mouse model, thereby facilitating cartilage repair [43].

As another example of inflammation-modulatory properties, Liu et al. developed a transformable peptide nanoplatform (BP-FFVLK-DSGLDSM, BFD) designed to treat rheumatoid arthritis (RA) by targeting the NF-κB/IκBα signaling pathway. BFD was designed to self-assemble into nanoparticles in circulation and transform into nanofibers within immune cells upon recognizing phosphorylated IκBα, thereby blocking IκBα degradation and suppressing NF-κB activation. This approach aimed to overcome the limitations of single-cytokine-target therapies in RA. The BFD reduced pro-inflammatory mediator expression, alleviated joint swelling, and prevented cartilage and bone destruction in a CIA rat model. The peptide demonstrated favorable biocompatibility and low toxicity, offering a novel and effective strategy for managing inflammatory diseases through in situ molecular recognition and structural transformation [44].

Addressing diabetic wound repair, Lu et al. utilized a hyaluronic acid (HA)-based hydrogel loaded with the Ac2-26 (Ac) peptide, which exhibited good mechanical properties, self-healing ability, and strong adhesion. This hydrogel modulated macrophage polarization, exerted anti-inflammatory effects, enhanced collagen deposition, and promoted angiogenesis, ultimately demonstrating enhanced diabetic wound healing (Figure 4c,d) [45].

Finally, in a nerve injury context, Su et al. developed a co-assembled peptide hydrogel (PII/NGF/NapFFY@Gel) using SPPS. This hydrogel suppressed NF-κB pathway activation and limited M1 macrophage polarization, resulting in a favorable anti-inflammatory environment that enhanced motor function recovery following peripheral nerve injury [46].

Taken together, these diverse studies underscore the versatility and therapeutic potential of AIP-functionalized hydrogels across a wide range of pathological conditions, including chronic wounds, autoimmune diseases, nerve injuries, and tissue degeneration. Despite varying design strategies and mechanisms of action—from cytokine inhibition and macrophage repolarization to NO sequestration and modulation of signaling pathways—the unifying theme lies in the localized, bioresponsive control of inflammation. These systems demonstrate the ability to respond to physiological stimuli such as pH, enzymes, and inflammatory cues, allowing for adaptive degradation, controlled drug or cytokine release, and immune microenvironment remodeling. Future research may focus on the integration of multiple anti-inflammatory mechanisms within a single hydrogel platform, combining immuno-regulatory, antimicrobial, and pro-regenerative functionalities. Emphasis on spatiotemporal release kinetics, enhanced tissue-specific targeting, and long-term biocompatibility will be critical for clinical translation. Furthermore, scaling up these materials to reproducible, GMP-compliant formulations will be essential to advance their use in precision immunotherapy and regenerative medicine.

**Figure 4 biomimetics-10-00442-f004:**
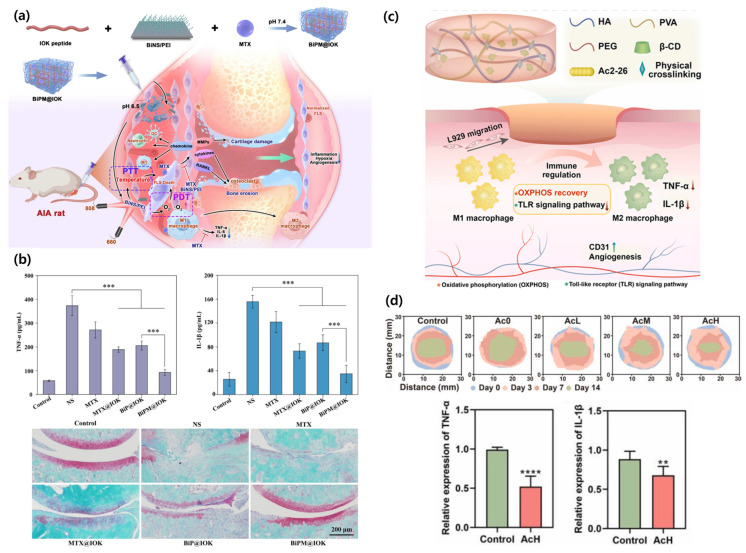
(**a**) Schematic representation of the systematic treatment strategy using BiPM@IOK hydrogel, which combines anti-inflammatory activity with targeted delivery for enhanced rheumatoid arthritis (RA) therapy. (**b**) Quantification of TNF-α and IL-1β of synovium tissue and Safranin O staining of joint sections after treatment with BiPM@IOK hydrogel (*** *p* < 0.001) [40]. (**c**) Schematic representation of the multifunctional hydrogel containing a polypeptide with anti-inflammatory and wound-healing function. (**d**) Representative images of the wound-healing progression over the treatment and relative gene expression of TNF-α and IL-1β (One-way ANOVA, ** *p* < 0.01 and **** *p* < 0.0001) [45].

The key anti-inflammatory properties of self-assembling peptides are summarized in Table 1.

### 2.2. Antimicrobial

#### 2.2.1. Mechanism of Antimicrobial Peptides (AMPs)

Antimicrobial peptides (AMPs) are small, amphipathic peptides that are widely distributed in nature and serve as key components of innate immunity in nearly all living organisms, including mammals, plants, and certain microorganisms. AMPs exhibit broad-spectrum antimicrobial activity against bacteria, fungi, viruses, and even antibiotic-resistant strains, primarily by interacting with and disrupting microbial membranes. Their structure typically features a balance of cationic and hydrophobic residues, enabling selective targeting of the negatively charged membranes of microbes while sparing host cells. Their potent activity and low tendency to induce resistance make them promising therapeutic candidates for the treatment of drug-resistant infections and represent a significant focus of medical research (Figure 5) [47,48].

The canonical mechanism of action for AMPs involves disruption of microbial membrane integrity. Upon initial electrostatic attraction to the bacterial surface, peptides orient themselves to facilitate insertion into the lipid bilayer. This interaction can be manifested through several well-established models. The barrel-stave model describes the insertion of α-helical peptides into the membrane to form transmembrane pores, while the carpet model suggests a mechanism akin to surfactant activity, where peptides accumulate parallel to the membrane until a threshold is reached, leading to membrane solubilization. The toroidal pore model, on the other hand, proposes a curvature-inducing mechanism wherein peptides promote pore formation that incorporates both peptide and lipid components in a continuous loop [49].

Beyond these classical frameworks, recent advances have brought attention to a class of AMPs: those capable of self-assembly into supramolecular structures. These SAPs harness non-covalent interactions to spontaneously organize into ordered architectures, including nanofibers, micelles, nanotubes, and hydrogels. These structural configurations often mimic biological assemblies, conferring enhanced functional properties such as increased proteolytic resistance, sustained antimicrobial activity, and targeted delivery in response to specific environmental stimuli (e.g., pH shifts or enzymatic activity). The resulting nanostructures can act as scaffolds for multivalent interactions with microbial surfaces, thereby amplifying membrane disruption or internalization effects. Moreover, their stability in physiological environments and potential for controlled disassembly allow for prolonged bioactivity and reduced cytotoxicity—a notable limitation of many conventional AMPs [50,51].

The modularity of self-assembled systems offers exciting opportunities for engineering tailored antimicrobial platforms. By fine-tuning peptide sequences, researchers can direct assembly behavior, modulate antimicrobial potency, and even incorporate dual-functional motifs that combine bactericidal and immunomodulatory activities. For instance, certain SAPs have demonstrated the capacity not only to eradicate pathogens but also to modulate inflammatory responses or recruit immune cells to the site of infection, thus linking immune defense with therapy [52].

In summary, while traditional AMPs act primarily through direct perturbation of microbial membranes, SAPs represent a next-generation strategy that integrates structural versatility with functional enhancement. Their biologically responsive, architecturally programmable nature positions them as formidable candidates in the ongoing pursuit of alternatives to conventional antibiotics, particularly in the context of rising antimicrobial resistance [53].

#### 2.2.2. Peptides for Enhancing Antimicrobial Properties in Hydrogels

AMPs are small, naturally occurring molecules with broad-spectrum activity against bacteria, fungi, and viruses, including drug-resistant strains, primarily by disrupting microbial membranes. Their amphipathic structures allow selective interaction with negatively charged microbial surfaces. Recently, self-assembling AMPs (SAMPs) have emerged, forming nanostructures such as fibers and hydrogels that enhance stability, bioactivity, and targeted delivery. SAMPs also enable functional customization, potentially combining antimicrobial and immunomodulatory functions, making them promising candidates in the fight against antimicrobial resistance.

Based on these mechanisms, Zhang et al. introduced an injectable, self-healing antimicrobial hydrogel composed of quaternized chitosan (QCS) and aldolized hyaluronic acid (AHA), integrated with self-assembling peptide nanofibers (PNFs) and ultrasmall silver nanoparticles (AgNPs). The authors aimed to address antibiotic resistance and infection control by combining broad-spectrum antimicrobial activity with high biocompatibility. The resulting QCS/AHA/PNF/AgNPs hydrogel exhibited excellent rheological properties, including shear-thinning and self-healing behavior, enabling precise application to irregular wound sites. It demonstrated >99.9% antibacterial efficacy against *E. coli* and *S. aureus*, along with no significant cytotoxicity toward human umbilical vein endothelial cells. This multifunctional hydrogel offers a promising strategy for treating infectious wounds and inspires the design of bioactive materials for broader biomedical applications (Figure 6) [54].

In an advanced investigation, Bundel et al. designed nanohydroxyapatite-loaded antimicrobial tripeptide gels to mimic the bone extracellular matrix and promote bone regeneration. Self-assembled tripeptides (Fmoc-FRF and Dpha-FRF) that form nanofibrous gels encapsulate nanohydroxyapatite, aiming to combine mechanical strength with antimicrobial function. The gels showed antimicrobial effects against *S. aureus* and promoted stem cell-driven osteogenic differentiation, indicating potential for bone tissue engineering [55].

Meanwhile, Eliza et al. reported a self-assembling peptide-based vaccine antigen, Q11-EH, targeting Gram-positive ESKAPE pathogens, including vancomycin-resistant *Enterococcus faecium* and methicillin-resistant *Staphylococcus aureus*. It featured a multi-presenting epitope (EH-motif) derived from the AdcA protein of *E. faecium*, utilizing the self-assembling Q11 peptide to enhance immunogenicity. The Q11-EH construct formed stable nanofibers and elicited antibodies capable of opsonophagocytic killing of multiple pathogens, demonstrating strong cross-protective potential in both passive and active immunization models. The opsonophagocytic inhibition assay confirmed the specificity of antibody binding to the EH epitope. This approach provides a promising strategy for developing broad-spectrum vaccines against nosocomial Gram-positive infections [56].

Moreover, Xu et al. discussed multifunctional PGK-CuTN coatings on titanium substrates for the treatment of infected bone defects. They developed a “sandwich”-structured coating by integrating graphene oxide, copper, and the antimicrobial peptide KR12 onto titanium dioxide nanorods using alkali-thermal treatment, dopamine self-polymerization, and layer-by-layer assembly. This coating exhibited excellent mechanical strength, hydrophilicity, and corrosion resistance while demonstrating significant antibacterial activity against *E. coli* and *S. aureus*. PGK-CuTN promoted M2 macrophage polarization, enhanced angiogenesis via HUVEC proliferation and VEGF/CD31 expression, and stimulated osteogenic differentiation of MC3T3-E1 cells. In vivo, the coating facilitated bone regeneration and reduced inflammation in osteomyelitis and bone defect models. This strategy offers a promising immune-modulatory and antibacterial-osteogenic platform for bone repair applications (Figure 7a,b) [57].

Extending the antimicrobial utility of SAPs, Zhou et al. used a peptide hydrogel formed from the natural antimicrobial peptide Jelleine-1 (PFKLSLHL-NH_2_). The self-assembled nanofibrous hydrogel was prepared without synthetic polymers or chemical cross-linkers, aiming to create an eco-friendly and biocompatible wound dressing. The hydrogel is injectable, self-healing, and responsive to pH and temperature changes. The results demonstrated strong antimicrobial activity, high biocompatibility, and effective wound healing in MRSA-infected burn models (Figure 7) [58].

To improve the stability of the Jelleine-1-based hydrogel, Zhang et al. designed the D-J-1 hydrogel, derived from the D-enantiomer of the antimicrobial peptide Jelleine-1. It self-assembles through hydrogen bonding, hydrophobic interactions, and π–π stacking forces. The D-J-1 hydrogel addresses the typical biodegradability issues of peptide hydrogels composed of L-amino acids by incorporating D-amino acids, enhancing enzymatic stability and bioavailability. This hydrogel exhibits excellent hemocompatibility due to its low hemolysis rate and significantly improves hemostatic performance compared to conventional gauze. Additionally, it effectively inhibits the growth of *Escherichia coli* and demonstrates anti-adhesion properties in a murine model, suggesting its potential as a promising alternative to current hemostatic agents and wound dressings [59].

A shear-sensitive hydrogel utilizing the KLVFF self-healing nano-fibrillar peptide, which originates from the Alzheimer’s amyloid-β (Aβ) sequence, was reported by Wiita et al. This KLVFF peptide demonstrated that at a concentration of 2.8 mM, it exhibits amphiphilic characteristics that enable potent antimicrobial activity by disrupting bacterial membranes. Furthermore, biocompatibility was confirmed via HEK-293 cell assays, showing no significant cytotoxicity. These properties suggest its potential application in preventing microbial infections and promoting wound healing [60].

Also, Liu et al. proposed a dual-action therapeutic strategy against antibiotic-resistant pathogens by developing an injectable antimicrobial hydrogel (C12G2), engineered through the self-assembly of antimicrobial peptides (AMPs). This hydrogel forms a supramolecular nanofiber network with shear-thinning and reversible properties, enabling precise delivery to infected sites while maintaining structural stability. In vivo experiments demonstrated its efficacy in treating MRSA-infected skin abscesses, matching the performance of vancomycin in eradicating infections and preventing abscess formation. Beyond direct antimicrobial activity, the C12G2 hydrogel exhibits immunoregulatory and anti-inflammatory effects, enhancing wound healing by modulating the immune response. Its mechanism combines membrane disruption of pathogens with cascade immune regulation, minimizing cytotoxicity and side effects [61].

Furthermore, Zhu et al. reported self-assembling nanopeptides, F3FT and N3FT, designed to combat intracellular bacterial infections and antimicrobial resistance. Engineered peptides by chimerizing a self-assembling core, a hydrophobic motif, and a cell-penetrating unit to achieve dual functionality: membrane penetration and antibacterial activity. The nanopeptides effectively entered host cells, eradicated intracellular *S. aureus*, and suppressed inflammation. The peptides disrupted bacterial membranes and induced ROS accumulation without significant toxicity. Moreover, no resistance emerged even after 30 days of bacterial exposure. This work highlights the therapeutic promise of multifunctional domain design in addressing intracellular infections and antibiotic resistance [62].

Collectively, these studies highlight the evolution of self-assembling antimicrobial peptides (SAPs) from simple membrane-disruptive agents into multifunctional therapeutic platforms with broad applications in tissue regeneration, hemostasis, immunomodulation, and targeted drug delivery. Moreover, SAPs demonstrate efficacy against drug-resistant and intracellular pathogens while minimizing cytotoxicity and immune overactivation. Their ability to integrate antimicrobial activity with immune regulation and tissue repair mechanisms makes them promising candidates for addressing complex clinical challenges such as chronic infections, wound healing, and infected bone defects.

Future advancements are expected to focus on optimizing molecular architecture for tunable degradation, maximizing therapeutic selectivity, and enabling spatiotemporal control of bioactive release. Integration with advanced delivery systems and scalable GMP-compliant manufacturing processes will be critical to translating SAP-based hydrogels into precision medicine and personalized therapeutic interventions.

**Figure 7 biomimetics-10-00442-f007:**
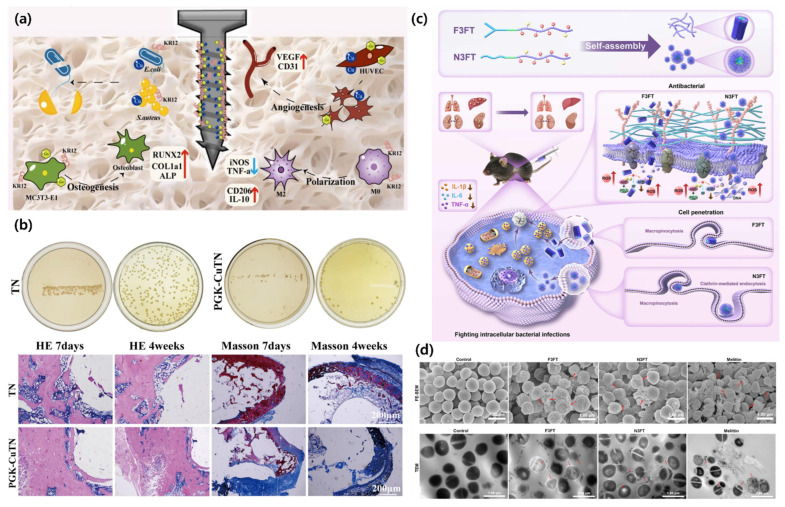
(**a**) Schematic illustration of the PGK-CuTN coating on a titanium substrate using multifunctional SAPs containing antimicrobial, angiogenesis, anti-inflammatory, and osteogenesis. (**b**) Representative images of TN and PGK-CuTN samples after 12 h of bacterial colony rolling cultivation, and histological analysis of bone defect implants via H&E and Masson’s trichrome staining [57]. Copyright © 2025, Elsevier. (**c**) Design and functional evaluation of self-assembling nanopeptides (F3FT and N3FT) with dual capabilities of cell penetration and antibacterial activity. (**d**) Representative FE-SEM and TEM images showing the membrane-disruptive effects of F3FT and N3FT on *S. aureus* [62].

The main antimicrobial properties of self-assembling peptides are summarized in Table 2.

### 2.3. Anticancer

#### 2.3.1. Mechanism of Anticancer Peptides (ACPs)

Anticancer peptides (ACPs) are a promising class of therapeutic agents capable of selectively targeting and eliminating cancer cells while sparing healthy tissues. These peptides, composed of short amino acid sequences, have garnered significant interest due to their structural versatility, biological specificity, and relatively low toxicity. Recent advancements have introduced SAP systems as an innovative extension of this field, offering enhanced stability, targeted delivery, and controlled activation in response to tumor-specific stimuli.

One of the key mechanisms by which ACPs exert their anti-tumor effects is through selective interaction with the cancer cell membrane. Compared to normal cells, cancer cells exhibit distinct changes in membrane composition, including externalization of phosphatidylserine, overexpression of O-glycosylated mucins, and an abundance of negatively charged sialic acids. These alterations result in a net negative surface charge, attracting cationic ACPs via electrostatic interactions. This selective affinity enables peptides to differentiate malignant cells from their healthy counterparts, making them highly valuable for targeted therapies [63].

Upon binding to the cancer cell membrane, many ACPs exert cytotoxic effects by disrupting membrane integrity. This process can occur through several biophysical models: the barrel-stave model, where peptides insert into the membrane forming transmembrane channels; the carpet model, where peptides align along the membrane surface creating a detergent-like effect; and the toroidal pore model, where the membrane curves inward to form continuous pores. These mechanisms ultimately cause leakage of cellular contents, ionic imbalance, and cell lysis. In many cases, this direct membrane disruption leads to rapid cell death without involving complex signaling pathways, allowing ACPs to circumvent common drug resistance mechanisms (Figure 8) [64].

In addition to their membranolytic activity, some ACPs penetrate the cell membrane and act on intracellular targets. Once internalized, they can localize mitochondria and induce the release of cytochrome c, activating caspase enzymes and triggering apoptosis via the intrinsic pathway. Other ACPs may interfere with crucial oncogenic signaling pathways, including PI3K/Akt, MAPK, and NF-κB, which are frequently dysregulated in various cancers. By inhibiting these pathways, ACPs suppress tumor proliferation, promote apoptosis, and reduce metastatic potential. Additionally, the nanostructured forms improve circulation time, enhance tumor accumulation via the enhanced permeability and retention (EPR) effect, and facilitate cellular uptake [65].

A particularly valuable characteristic of ACPs is their responsiveness to the tumor microenvironment. These systems can be engineered to remain inactive during systemic circulation and activate only under specific tumor-associated conditions. For instance, changes in pH, the presence of matrix metalloproteinases, or elevated reactive oxygen species can trigger structural transformations in the peptide assembly, exposing the active domain and initiating cytotoxic action. This targeted activation reduces off-target effects and enhances therapeutic precision, making self-assembling ACPs (SACPs) highly suitable for in vivo applications [66,67].

Moreover, SACPs can serve as multifunctional platforms for the co-delivery of anticancer drugs, genetic material, or imaging agents. In this capacity, they function not only as therapeutic agents but also as carriers, enabling combination therapy and real-time tumor monitoring. Co-assembly of ACPs with chemotherapeutic agents such as doxorubicin or cisplatin can produce synergistic effects and enhance tumor cell death while minimizing systemic toxicity. Some designs incorporate stimuli-responsive elements that allow controlled release of cargo molecules within the tumor microenvironment, further improving therapeutic outcomes [68].

Beyond their direct anti-tumor activity, ACPs have also been shown to modulate the immune response. Certain peptides can activate dendritic cells, stimulate the production of pro-inflammatory cytokines, or enhance the cytotoxic activity of T cells and natural killer cells. This immunomodulatory potential allows ACPs to act as both direct cytotoxic agents and immune adjuvants, contributing to a more comprehensive and sustained anticancer response. Additionally, some ACPs inhibit angiogenesis, a critical process in tumor growth and metastasis, by blocking vascular endothelial growth factor (VEGF) signaling pathways and disrupting new blood vessel formation [69].

An important advantage of peptide-based therapeutics is their inherent biocompatibility and biodegradability. Unlike many synthetic anticancer agents that can accumulate in tissues and cause long-term side effects, ACPs are naturally broken down into amino acids by enzymatic processes after fulfilling their therapeutic function. This feature reduces the risk of chronic toxicity and makes these peptides more favorable for long-term clinical application [70].

In conclusion, anticancer peptides represent a diverse and adaptable class of therapeutic molecules capable of engaging multiple cellular targets and mechanisms. The development of SACPs has significantly expanded their therapeutic potential by enhancing peptide stability, improving tumor specificity, and enabling stimulus-responsive delivery. As research advances, SAP systems are expected to play an increasingly important role in precision oncology, offering safer, more effective, and personalized cancer treatments.

#### 2.3.2. Peptides for Enhancing Anticancer Properties in Hydrogels

SACPs are emerging as therapeutic agents that selectively target cancer cells while sparing healthy tissues. These peptides further enhance this potential by offering improved stability, targeted delivery, and stimulus-responsive activation within the tumor microenvironment. SACPs typically bind to negatively charged cancer cell membranes, disrupt membrane integrity, or penetrate cells to induce apoptosis by targeting mitochondria or oncogenic signaling pathways. SACPs can be engineered to activate only under tumor-specific conditions, reducing side effects and increasing precision. Additionally, they enable co-delivery of drugs, genes, or imaging agents, making them highly promising tools in precision oncology.

Tumor microenvironment (TME) exhibits distinct physiological and biochemical characteristics compared to healthy tissues, such as low pH, elevated levels of reactive oxygen species (ROS), and increased activity of specific enzymes such as matrix metalloproteinase (MMPs) and other proteolytic enzymes. These unique characteristics of the TME have been actively exploited in the design of responsive therapeutic systems. To use the acidic condition of the TME, Wu et al. developed a tumor microenvironment (TME) pH-responsive multilevel self-assembling peptide, Ac–AAAFFHH–NH_2_. At physiological pH (7.4), the peptide formed large aggregates (~1.56 μm in size), whereas under mildly acidic conditions (pH 6.4), mimicking the TME, it transformed into nanomicelles (<100 nm) that could efficiently internalize tumor cells such as HeLa, MCF-7, and 4T1. This pH-triggered structural transition enabled selective cellular uptake and induced apoptosis via tumor cell cytotoxicity (Figure 9) [71].

Other tumor-specific biochemical cues, such as redox potential, have also been utilized. Mei et al. developed the Pep-CS-LND hydrogel, designed to deliver anticancer drugs specifically to mitochondria, which are critical for cancer cell survival. This targeted approach may increase chemotherapy effectiveness while minimizing damage to healthy cells, addressing a major drawback of traditional treatments. The hydrogel employed a redox-responsive mechanism enabling controlled drug release in response to elevated glutathione (GSH) levels found in cancer cells. This specificity can lead to improved therapeutic outcomes by releasing drugs precisely where needed, thereby reducing systemic side effects [72].

Similarly, Zhang et al. developed the aP/IR@FMKB hydrogel, designed to release drugs in response to MMP-2 enzyme activity, which is commonly elevated in cancerous tissues. By conjugating doxorubicin with a trastuzumab epitope and an MMP-2-sensitive peptide linker (Fmoc-KPLGLAGCRGDK), the hydrogel is susceptible to degradation by MMP-2, allowing targeted drug release in the tumor microenvironment. This strategy improves targeting of cancer cells and holds potential for better treatment outcomes in patients with specific tumor types [73].

To improve intratumoral penetration and retention, Wang, Ying, et al. developed an octa-arginine (R8)-modified RADA16 (RR) self-assembling peptide nanofiber hydrogel, demonstrating its capabilities in selective tumor penetration, sustained drug release, and hemostasis. This peptide hydrogel enhanced cellular uptake, leading to apoptosis in SKOV3/MDR cells while inhibiting cell migration. Furthermore, in a mouse model, it significantly suppressed tumor growth and promoted intratumoral drug accumulation (Figure 10a,b) [74].

Expanding on intracellular targeting, Wu, Xia, et al. demonstrated that the glycopeptide selectively degraded GLUT1 (glucose transporter 1), which is overexpressed in most cancer cells, thereby exerting a tumor-suppressive effect in HCT-116 cells. The glycopeptide underwent lysosome-mediated self-assembly, resulting in reduced GLUT1 expression and subsequent increases in reactive oxygen species (ROS) and LC3-II levels, ultimately inducing apoptosis. These findings highlight the potential of this strategy for the development of targeted cancer therapies [75].

Therapeutic approaches that target or modulate the immune microenvironment have recently attracted attention, with immunotherapy and vaccines becoming a central pillar of modern cancer treatment. Expanding on the immunomodulatory ability of SACPs, Luo et al. designed two peptides, RLDI and RQDT, which significantly enhanced the biological functions of dendritic cells (DCs). These peptides facilitate sustained antigen release, promote DC adhesion, and improve DC maturation. As a result, the SAPs increased the ability of DCs to present antigens effectively, which is crucial for eliciting a robust immune response against tumors. The study demonstrated that peptide hydrogel-based DC treatments can achieve both preventive and therapeutic effects against colon cancer, suggesting that the incorporation of these peptides into DC vaccines could improve outcomes in cancer immunotherapy (Figure 10c,d) [76].

To further enhance localized immunotherapy, Guan et al. demonstrated several practical implications for cancer treatment using the N-Pep-Mn gel. This hydrogel demonstrates the ability to retain Mn^2+^ ions and αPD-1 antibodies at the tumor site for extended periods. Such localized retention minimizes systemic side effects commonly associated with traditional immunotherapy, which can lead to off-target toxicity. The hydrogel enables controlled and sustained release of Mn^2+^ ions and αPD-1 antibodies, which is crucial for maintaining effective therapeutic concentrations at the tumor site, thereby enhancing the immune response while reducing rapid drug metabolism and loss of efficacy [77].

**Figure 10 biomimetics-10-00442-f010:**
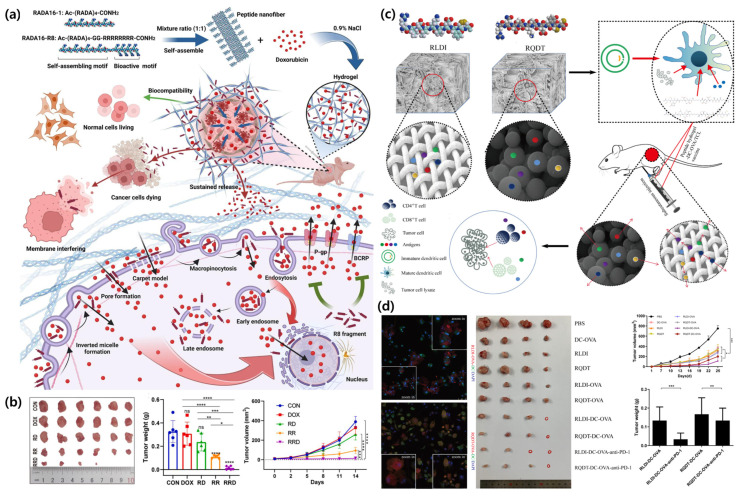
(**a**) Schematic illustration of the preparation of octa-arginine (R8)-modified RADA16 (RR) SAP nanofiber hydrogel and its mechanisms and therapeutic benefits as a doxorubicin (DOX) delivery system for anticancer therapy. (**b**) Representative images of dissected tumors, quantification of tumor weights, and tumor growth curves to evaluate the in vivo anticancer effect of SAPs (One-way ANOVA, ‘ns’ means not significant. * *p* < 0.05, ** *p* < 0.01, *** *p* < 0.001, and **** *p* < 0.0001) [74]. (**c**) Proposed model illustrating SAPs as a novel vaccine delivery platform that forms nanostructures to load various vaccine types, support dendritic cell (DC) activation, enable sustained antigen release, and enhance CD^4+^/CD^8+^ T cell responses for effective anti-tumor immunity. (**d**) Confocal microscopy imaging showing DC recruitment stimulated by RLDI- or RQDT-OVA treatment, along with representative images of dissected tumors; the quantification of tumor weight and volume demonstrating the therapeutic efficacy of peptide hydrogel-loaded DC vaccines (One-way ANOVA, ‘ns’ means not significant. * *p* < 0.05, ** *p* < 0.01, and *** *p* < 0.001) [76].

Recognizing the limitation posed by poorly immunogenic tumors, Wang, He, et al. designed TEP-FFG-CRApY, which effectively penetrates prostate cancer (PCa) and induces ferroptosis, thereby inhibiting tumor growth. This SACP assembles into nanoparticles and transforms into nanofiber in response to alkaline phosphatase (ALP) in tumor cells, promoting lysosomal membrane permeabilization and ferroptosis, thus enhancing the immunogenicity of immunologically “cold” tumors, ultimately contributing to prolonged survival [78].

In conclusion, the distinct characteristics of the TME have been successfully leveraged to develop responsive SACP-based therapies. These systems allow for targeted drug delivery and controlled release, thereby enhancing therapeutic outcomes. Furthermore, SAPs with immunomodulatory properties offer promising strategies for improving anti-tumor immunity and overcoming the limitations of conventional immunotherapy.

Future advancements are anticipated in the clinical translation of these systems, particularly through the integration of multifunctional therapeutics and diagnostics to enable personalized cancer treatment strategies.

The principal anticancer properties of self-assembling peptides are summarized in Table 3.

### 2.4. Bioimaging

#### 2.4.1. Mechanism of Bioimaging Peptides

In the field of molecular imaging, peptide-based probes have become increasingly important due to their favorable biological properties. Peptides are inherently biocompatible, easily modifiable, and capable of selectively targeting specific biomolecules. Among various peptide-based tools, self-assembling peptides (SAPs) have garnered particular attention for their ability to form stable nanostructures and perform complex functions in bioimaging applications. These self-assembling systems can be engineered to interact precisely with biological environments, enabling improved visualization of cellular and molecular processes.

In bioimaging, SAPs offer several unique advantages. First, the nanostructures formed through self-assembly often exhibit enhanced stability and prolonged retention at target sites compared to unstructured peptides, resulting in improved imaging contrast and an extended imaging window. Second, SAPs present multiple functional groups on their surfaces, allowing them to carry higher payloads of imaging agents such as fluorescent dyes or contrast enhancers. This multivalency produces stronger, more localized signals, thereby increasing imaging sensitivity and resolution (Figure 11) [79,80].

One of the most common imaging techniques employing SAPs is fluorescence imaging, where peptides are tagged with fluorescent molecules that emit light upon excitation at specific wavelengths. Some SAPs are designed to exhibit aggregation-induced emission (AIE), whereby peptides remain non-fluorescent in monomeric form but become highly fluorescent upon aggregation. This property is particularly useful for detecting assembled structures selectively in target areas such as tumor tissues, where peptides accumulate and self-assemble [81].

Magnetic resonance imaging (MRI) also benefits from SAPs, which can be functionalized with gadolinium ions or iron oxide nanoparticles as contrast agents. Clustering of these agents within self-assembled nanostructures significantly enhances local magnetic signals, resulting in clearer, more precise MRI images and better differentiation between healthy and diseased tissues. In positron emission tomography (PET) and single-photon emission computed tomography (SPECT), peptides labeled with radioisotopes such as copper-64 (^64^Cu) or technetium-99m (^99^mTc) emit detectable radiation. When carried by SAP structures, these radioactive elements concentrate signals in specific body regions, enabling real-time tracking of biological processes or disease progression [68].

Photoacoustic imaging is another advanced technique where SAPs contribute significantly. This method relies on chromophores incorporated into peptide structures absorbing light energy, which is then released as heat, generating acoustic waves captured by ultrasound detectors. This approach enables imaging at greater tissue depths with higher spatial resolution, enhanced by SAPs stabilizing chromophores and directing them to target tissues [82,83,84].

Besides signal enhancement, SAPs improve biological targeting. Many peptides include specific sequences that bind receptors on diseased cells. For example, the RGD motif binds integrin receptors, commonly overexpressed on tumor cells and tumor-associated vasculature. Upon binding, peptide assemblies undergo receptor-mediated endocytosis. Some nanostructures are engineered to disassemble intracellularly, releasing imaging agents only after uptake, thereby reducing off-target effects and increasing imaging accuracy.

Another key advantage of SAPs in bioimaging is their biodegradability. Unlike many synthetic nanoparticles, peptides are enzymatically degraded into amino acids or small fragments, which are non-toxic and easily metabolized or excreted. This enhances safety, particularly for repeated or long-term imaging studies. From a biological perspective, interactions between SAPs and living systems are highly controllable. Peptides can be designed to respond to disease-specific signals, ensuring imaging occurs only under desired conditions. This responsive behavior enables highly selective imaging, reducing background noise and increasing diagnostic accuracy [85,86,87].

In summary, SAPs represent a significant advancement in designing intelligent imaging agents. Their ability to self-organize, respond to biological stimuli, and deliver imaging payloads to specific targets makes them ideal candidates for precise, non-invasive visualization of complex biological systems. Ongoing research is expected to expand their role in diagnostic and therapeutic applications.

#### 2.4.2. Peptides for Enhancing Bioimaging Properties in Hydrogels

SAPs have emerged as promising molecular imaging tools due to their biocompatibility, structural stability, and target specificity. They form nanostructures that enhance imaging contrast, prolong retention at target sites, and deliver higher loads of imaging agents such as fluorescent dyes or radioisotopes. SAPs support multiple imaging modalities, including fluorescence, MRI, PET/SPECT, and photoacoustic imaging, while enabling responsive, disease-specific targeting. Their biodegradability and low toxicity further improve safety for in vivo applications, offering a versatile platform for non-invasive, real-time biological visualization.

Conventional molecular imaging tools are often limited by short penetration depth. To improve deep tissue penetration, Chu, Yang, et al. developed rigid α-helical polypeptide nanoprobes (_L_-PTN) with thermally activated delayed fluorescence (TADF) for time-resolved, high-contrast bioimaging. _L_-PTN integrated TADF fluorophores covalently onto α-helical polypeptide scaffolds, providing orange emission (>600 nm) and microsecond-scale fluorescence lifetimes. The α-helical structure suppresses TADF molecule rotation and vibration and minimizes aggregation-caused quenching, enhancing fluorescence intensity and lifetime. This peptide showed deep tissue penetration with eliminated autofluorescence (Figure 12) [88].

Building on the need for both diagnostic precision and therapeutic intervention, Guo, Yuanyuan, et al. presented the development of an endothelium-targeted NF-κB siRNA nanogel (VP-Gd-NF-NG) for atherosclerosis diagnosis and treatment. This nanogel integrates NF-κB p65 subunit-targeting siRNA (siNF-κB), a VCAM-1 targeting peptide (VHSPNKK, VP) for inflamed endothelium, and a magnetic resonance (MR) contrast agent (Gd-DOPA), fabricated via click chemistry and complementary base-pairing. In vitro and in vivo studies demonstrated that VP-Gd-NF-NG suppresses p65 expression in inflamed endothelial cells, reducing cell adhesion molecules, chemokines, and inflammatory cytokines (IL-6, IL-8, and MCP-1), thereby decreasing monocyte accumulation and attenuating atherosclerotic plaques. In ApoE−/− mouse models, VP-Gd-NF-NG reduced plaque proportions from 34.5% to 21.5% and enabled real-time MR monitoring of treatment platforms (Figure 13a) [89].

While vascular targeting has shown promise, brain-target imaging presents unique challenges due to the blood–brain barrier (BBB). Addressing this, Li et al. highlighted the transformative potential of the Y-shaped neuropeptide Y (NPY)-mimetic peptide-dye self-assembly system designed to cross the BBB target glioma mitochondria and enable noninvasive NIR-II fluorescence imaging for diagnosis and therapeutic monitoring. The system incorporates chiral NPY-mimetic peptides (^D/L^NP9Y(14)) with an enterokinase (ENTK)-recognizable sequence, coupled with the NIR-II fluorophore IR1048. NIR-II fluorescence beyond 1300 nm offers deep tissue penetration and low autofluorescence interference, enabling high-resolution noninvasive imaging of brain tumors, which could be applied clinically for preoperative tumor boundary delineation or postoperative residual tumor monitoring. The self-assembly of ^D^NPY(14)-ENTK-IR1048 provided a sustained fluorescence signal for over 7 days, facilitating long-term tumor tracking and treatment response assessment [90].

Complementing fluorescence-based imaging, photoacoustic imaging has gained attention for its ability to provide deeper penetration and spatial resolution. Borum, Raina M., et al. presented noncovalent assemblies of iRGD peptides and methylene blue dyes for targeted cancer imaging and therapy. By tuning peptide sequences to enhance electrostatic or hydrophobic interactions, the authors created nanostructures with distinct photoacoustic properties. In particular, branched nanoparticles formed with iRGD-DD exhibited a redshifted absorbance peak at 720 nm, enabling strong photoacoustic imaging signals. These SAPs-based nanoparticles selectively targeted tumors, and upon irradiation, produced significant reactive oxygen species for photodynamic therapy [91].

Expanding the utility of NIR-II imaging further, Luan, Xin, et al. presented a 2D nanoplatform, peptide nanosheets (PNS)/PEG-Ag_2_S quantum dots (QDs) nanohybrids, designed for near-infrared-II (NIR-II) fluorescent bioimaging and photothermal therapy (PTT) of tumors. PNS were synthesized with uniform structures via SAPs, and PEG-functionalized Ag_2_S QDs were covalently linked to PNS through amide bonds. From a bioimaging perspective, PNS/PEF-Ag_2_S QDs offered deep tissue penetration and high fluorescence intensity in the NIR-II, enabling high-resolution visualization of tumor sites. Furthermore, PNS/PEG-Ag_2_S QDs had the potential to effectively convert light into heat for tumor destruction. This high efficiency can lead to better therapeutic outcomes. This nanoplatform demonstrated potential for precise tumor diagnosis and visual monitoring of PTT progress through NIR-II fluorescence [92].

The rationale for developing bioimaging technology by combining aggregation-induced emission luminogens (AIEgens) and peptides is to combine the excellent fluorescent properties of AIEgens with the biocompatibility and functionality of peptides to enable high-sensitivity, high-definition, and target-directed imaging. This conjugate overcomes the limitations of existing technologies and enables precise bioimaging in tumor diagnosis, cellular organelle visualization, disease mechanism studies, and more. For example, Hernando-Muñoz, Carla, et al. presented a novel methodology for creating programmable nanovesicles and soft nanocarriers by combining AIEgens with naturally derived depsipeptides. AIEgen-depsipeptide hydrides self-assemble via hydrogen bonding and π-π stacking interactions to form stable hollow nanovesicles, maintaining structural integrity in water or water-organic solvent mixtures. These nanovesicles exhibit red emission at 670 nm under 390 nm excitation, with AIEgen properties retention in A549 lung cancer cells, confirmed by confocal laser scanning microscopy and STED super-resolution microscopy, with vesicle sizes consistent with AFM and FESEM results. These nanovesicles delivered physiologically active peptides into cells, functioning as both biomarkers and nanocarriers, enabling high-contrast bioimaging and time-programmed drug delivery [93]. Taking the concept further into therapeutic monitoring, Pei, Shicheng, et al. underscored the potential of AIEgen-peptide nanoprobes to enhance cancer treatment precision by integrating bioimaging and chemotherapy. The nanoprobe comprised a TPE imaging motif, a Tyr-Tyr self-assembling motif, an acid-activable DOX prodrug, a caspase-3 cleavable DEVD linker, and an RGD targeting motif, self-assembling into ~187 nm spherical nanoparticles via hydrogen bonding and π-π stacking. In the tumor microenvironment at acidic pH, hydrazone bonds hydrolyze, releasing DOX, which induces apoptosis and activates caspase-3, cleaving the DEVD linker. This triggers TPE-1 to reassemble into AIE-active nanofibers, significantly increasing TPE fluorescence and the F_TPE/F_DOX ratio, enabling real-time monitoring of therapeutic efficacy. The nanoprobe demonstrated selective uptake via RGD-mediated integrin targeting, with high intracellular fluorescence signals. In vivo, TPE-1(Hyd-DOX)-DEVD-Cy5 nanoparticles accumulated in tumor sites over 48 h under 605 nm excitation, showing superior tumor suppression and biocompatibility compared to free DOX (Figure 13b,c) [94].

In conclusion, SAPs are emerging as powerful platforms in advanced molecular imaging and targeted therapy. The SAPs’ structural adaptability, biocompatibility, and ability to be combined with functional imaging moieties, such as TADF, NIR-II fluorophores, photoacoustic dyes, and AIEgens, allow for precise high-contrast visualization, prolonged retention, and drug delivery. These systems enhance targeting, improve imaging depth and clarity, and enable real-time monitoring of treatment. These peptides can self-assemble and respond to the TME, further supporting controlled drug delivery and therapeutic feedback. Collectively, these SAP-based systems represent a transformative step toward precision diagnostics and theranostics, with broad potential across cancer, cardiovascular, and neurological diseases.

The major antimicrobial properties of self-assembling peptides are summarized in Table 4.

## 3. Conclusions

SAPs applied in hydrogels are gaining attention in various biomedical fields—including anti-inflammatory, antimicrobial, anticancer, and bioimaging applications—due to their diverse biological functions. These peptides can effectively suppress inflammatory responses by inhibiting pro-inflammatory signaling pathways, alleviating oxidative stress, and protecting their structure from protein-degrading enzymes. Additionally, SAPs exhibit antimicrobial effects by selectively disrupting bacterial membranes or by forming nanostructures in response to biological stimuli, which contributes to addressing the existing problem of antibiotic resistance. Designed as stimulus-responsive systems that activate in the tumor microenvironment and release drugs, SAP hydrogels can be utilized for anticancer therapy. Moreover, by selectively binding to target tissues and leveraging their self-organizing properties, SAPs can serve as smart imaging platforms for high-resolution bioimaging. In this way, SAP hydrogels can dynamically control their structure and function in response to biological signals, making them adaptable to complex environments and capable of amplifying various biological signals. As a result, SAP-based hydrogel biomaterials are seeing expanding potential for use in anti-inflammatory, antimicrobial, and anticancer therapy, diagnostics, and other biomedical fields (Figure 14).

## 4. Challenges and Future Perspectives

Self-assembling peptides (SAPs) and related peptide-based biomaterials have garnered significant interest for their potential in biomedical applications. These materials offer tunable sequences, inherent biocompatibility, and the ability to form hierarchically organized nanostructures. However, several formidable obstacles must be overcome before SAP-based hydrogels can be broadly adopted in patient care and translational research.

One of the most pressing issues is the inherent limitation of current peptide synthesis technologies. As peptide chain length increases, the isolated yield declines exponentially due to cumulative inefficiencies in each coupling cycle during solid-phase peptide synthesis (SPPS). Small fractions of unreacted residues or deletion sequences accumulate step by step, becoming a dominant yield-limiting factor for long sequences. Sequence-dependent aggregation and secondary structure formation on the resin further impede reagent accessibility, compounding coupling inefficiency. Additionally, these synthesis-related challenges are exacerbated by unavoidable losses during purification processes. Long-chain peptides typically require multiple rounds of high-performance liquid chromatography (HPLC) or other chromatographic steps, each of which diminishes overall recovery [95,96,97].

To address these synthetic bottlenecks, alternative strategies have been developed in parallel: segment condensation, native chemical ligation, and enzymatic conjugation all assemble shorter, high-purity peptide fragments in solution rather than relying on a single, linear SPPS process. This modular approach minimizes the cumulative coupling losses inherent to long, uninterrupted sequences. In addition, approaches such as microwave-assisted coupling, continuous-flow SPPS, and the adoption of environmentally friendly solvents and recyclable resins have further improved coupling efficiency and sustainability [98,99]. For industrial-scale production, an alternative strategy may involve first optimizing peptide sequences via SPPS at a small scale and then transitioning to biosynthetic production systems for cost-effective and scalable manufacturing.

Beyond synthesis, SAPs face significant biological challenges that restrict their therapeutic utility. Most notably, SAPs are highly susceptible to enzymatic degradation in vivo, which leads to rapid clearance and poor bioavailability. For example, many linear therapeutic peptides exhibit plasma half-lives of merely minutes to a few hours, necessitating frequent dosing and often rendering oral administration impractical. To improve in vivo stability, strategies such as peptide cyclization, backbone stapling, PEGylation, and incorporation of albumin-binding domains have been explored. While these modifications can substantially improve pharmacokinetics, they also increase synthetic complexity and must be rigorously evaluated for potential immunogenicity [100,101]. Therefore, future SAP research should prioritize the use of supramolecular self-assembly strategies to achieve biological stability without relying solely on complex chemical modifications. Future SAP designs should incorporate structural motifs that enhance enzymatic resistance through stable secondary structures and steric shielding, thereby creating intrinsically degradation-resistant supramolecular architectures.

Immunological safety remains a critical consideration in clinical development. Although peptides generally provoke fewer immune responses than full-length proteins, non-natural amino acids, trace impurities, or aggregated species can still elicit unwanted immunogenicity. Regulatory agencies now impose stringent requirements for impurity profiling, aggregate control, and product homogeneity. Moreover, the regulatory classification of peptides varies by region: in the United States, the FDA typically treats synthetic peptides of 40 residues or fewer as chemical drugs, whereas longer chains or recombinant products may be regulated as biologics. Similarly, in the European Union, EMA guidance often parallels FDA regulations but may impose varying length thresholds and documentation standards, creating additional complexity for global development programs.

Another translational barrier is the limited scope of current biological evaluations. Most reported studies have been restricted to in vitro analyses or small animal models. For clinical translation, comprehensive evaluation in large animal models—including assessments of toxicity, immunogenicity, and pharmacokinetics—is essential. Moreover, early consideration of regulatory classification—whether as a medical device, combination product, or therapeutic—will be crucial for streamlining future approval pathways.

Scale-up manufacturing presents yet another formidable challenge. While SPPS is well-suited to small-scale laboratory synthesis, its extension to commercial production suffers from low overall yields and high solvent consumption. Large-scale manufacture of peptide therapeutics—such as GLP-1 analogues—can demand hundreds of kilograms of peptide per year, consuming vast quantities of dimethylformamide (DMF), trifluoroacetic acid (TFA), and other organic solvents and generating correspondingly large volumes of chemical waste. These factors drive up production costs and raise serious environmental sustainability concerns. Accordingly, the development of greener synthesis methods—solvent recycling systems, alternative solvent platforms, and resin-reuse protocols—is essential to ensure economically viable and environmentally responsible industrial peptide production [102,103,104,105].

Looking forward, the integration of advanced peptide synthesis, nanotechnology-based delivery systems, and systems-level biological insights will be indispensable for advancing SAP-based hydrogel platforms. To achieve clinically relevant performance, future SAPs must balance molecular precision, biofunctionality, and manufacturability. A multidisciplinary approach that brings together peptide engineering, materials science, pharmacology, and regulatory expertise—supported by strong collaboration across academia, industry, and regulatory bodies—will be critical in unlocking the full clinical potential of SAP-based hydrogels in regenerative medicine, targeted therapy, and biomedical diagnostics.

## Figures and Tables

**Figure 1 biomimetics-10-00442-f001:**
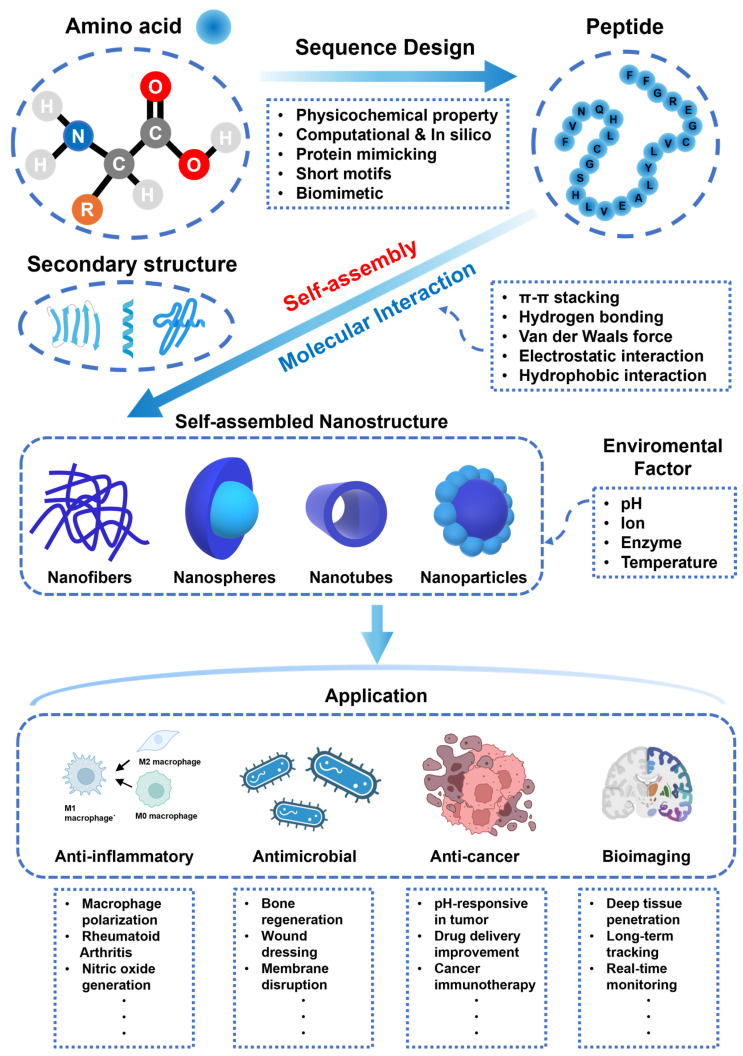
Schematic illustration of the overall self-assembly process and biomedical applications of self-assembling peptides.

**Figure 2 biomimetics-10-00442-f002:**
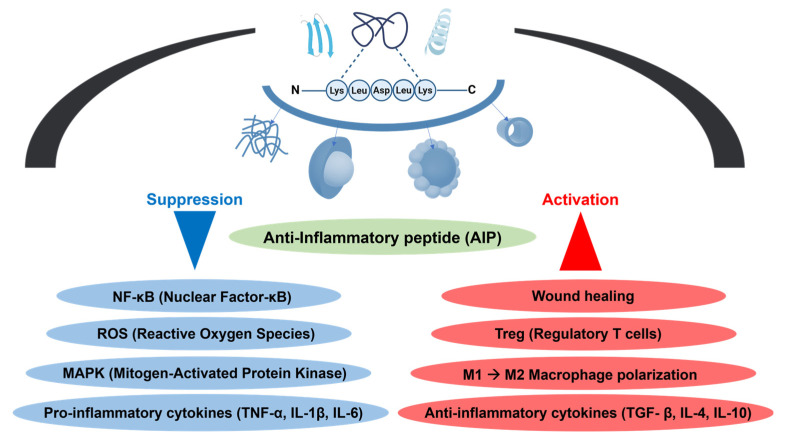
Illustration of anti-inflammatory peptide-mediated regulation of key immune components and inflammatory pathways.

**Figure 3 biomimetics-10-00442-f003:**
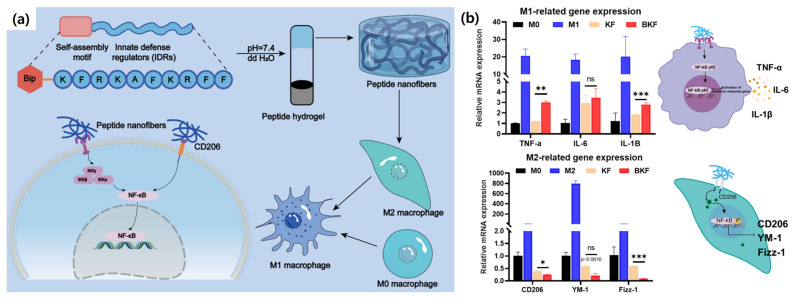
(**a**) Schematic of the SAP nanofiber-induced macrophage repolarization pathway. (**b**) qPCR analysis of M1- and M2-associated gene expression in macrophage treatment with SAPs (One-way ANOVA, * *p* < 0.05, ** *p* < 0.01, and *** *p* < 0.001) [38].

**Figure 5 biomimetics-10-00442-f005:**
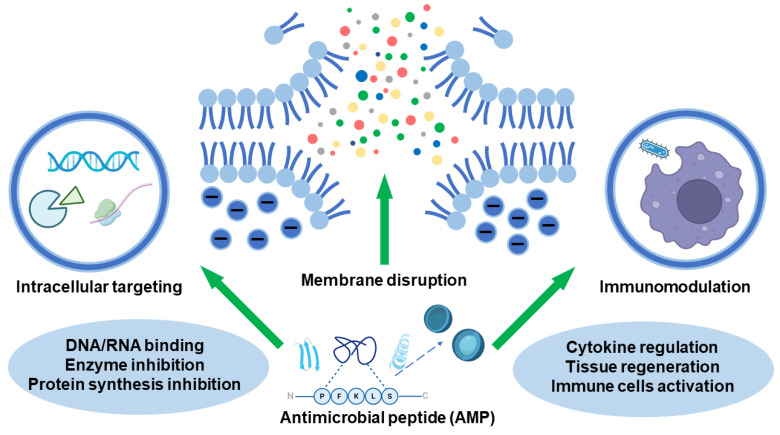
Overview of antimicrobial modes of action of antimicrobial peptides.

**Figure 6 biomimetics-10-00442-f006:**
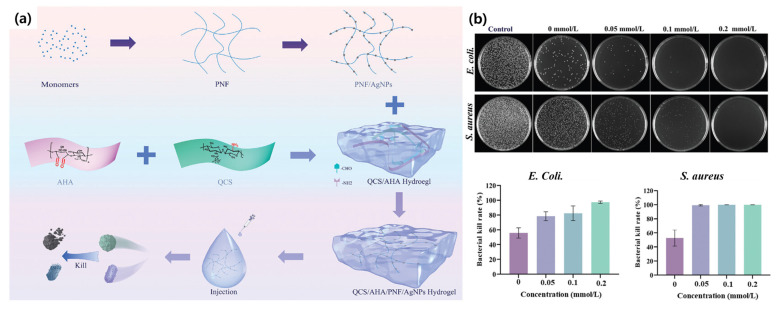
Schematic showing the synthesis of an antimicrobial hydrogel composed of quaternized chitosan (QCS), aldolized hyaluronic acid (AHA), peptide nanofibers (PNFs), and silver nanoparticles (AgNPs) (**a**), with its antibacterial properties evaluated through the effects of varying PNF/AgNPs content on *E.coli* and *S. aureus*, along with corresponding statistical analyses (**b**) [54]. Copyright © 2024, John Wiley and Sons.

**Figure 8 biomimetics-10-00442-f008:**
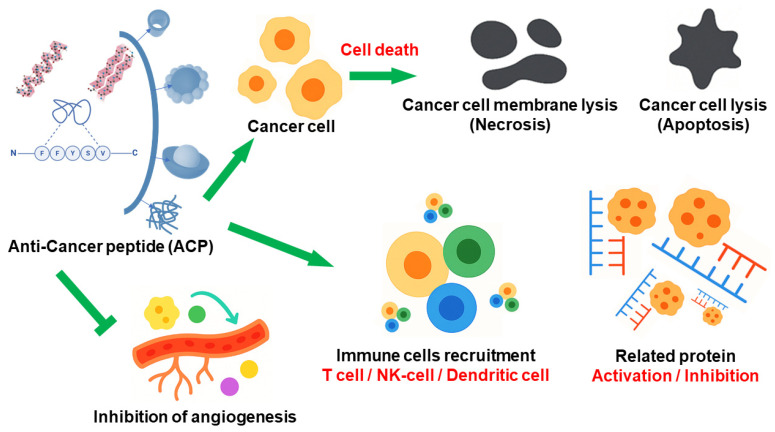
Schematic representation of anticancer peptide-induced cell death, inhibition of angiogenesis, and modulation of immune components.

**Figure 9 biomimetics-10-00442-f009:**
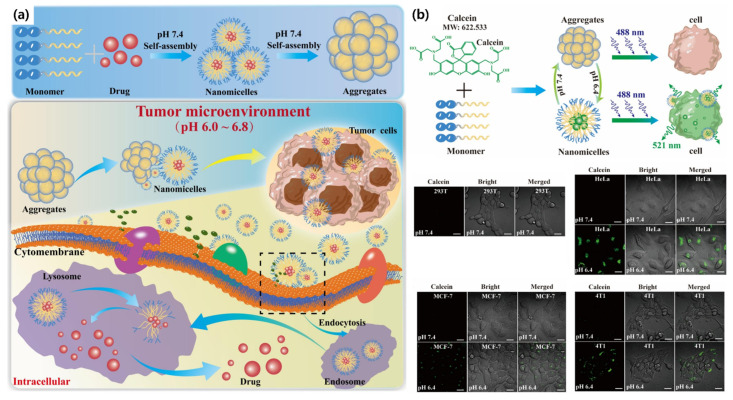
(**a**) Schematic illustration and functional evaluation of size-transformable peptide self-assembling particles (St-PSAPs) designed for tumor microenvironment (TME) pH-responsive, tumor-specific therapy. (**b**) Cellular uptake of St-PSAPs was assessed under pH 6.4 and pH 7.4 conditions across various cell types, demonstrating enhanced uptake in acidic TME. The scale bar is 20 μm [71]. Copyright © 2025, Elsevier.

**Figure 11 biomimetics-10-00442-f011:**
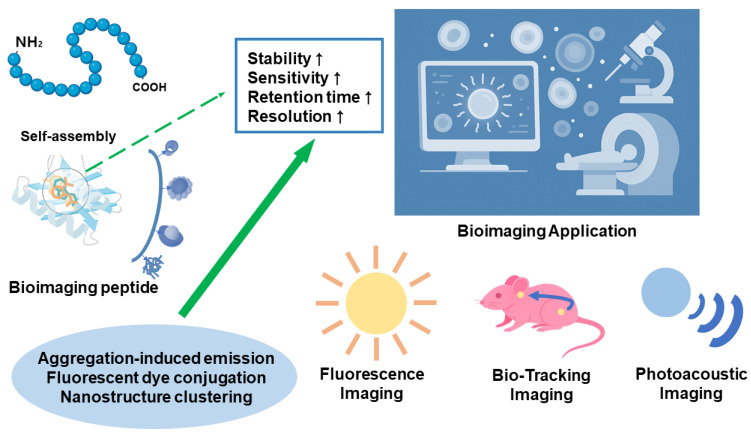
Comprehensive illustration of imaging capability enhancement facilitated by bioimaging peptides.

**Figure 12 biomimetics-10-00442-f012:**
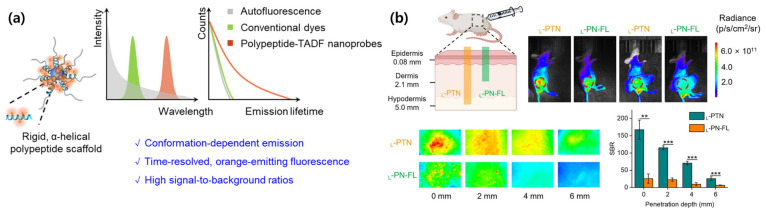
(**a**) Schematic illustration of the characterization and conformation-dependent photophysical properties of α-helical polypeptide nanoprobes. (**b**) Enhanced imaging contrast and deep tissue penetration achieved by α-helical polypeptide nanoprobes (Student’s *t*-test, ** *p* < 0.01 and *** *p* < 0.001) [88]. Copyright © 2025, American Chemical Society.

**Figure 13 biomimetics-10-00442-f013:**
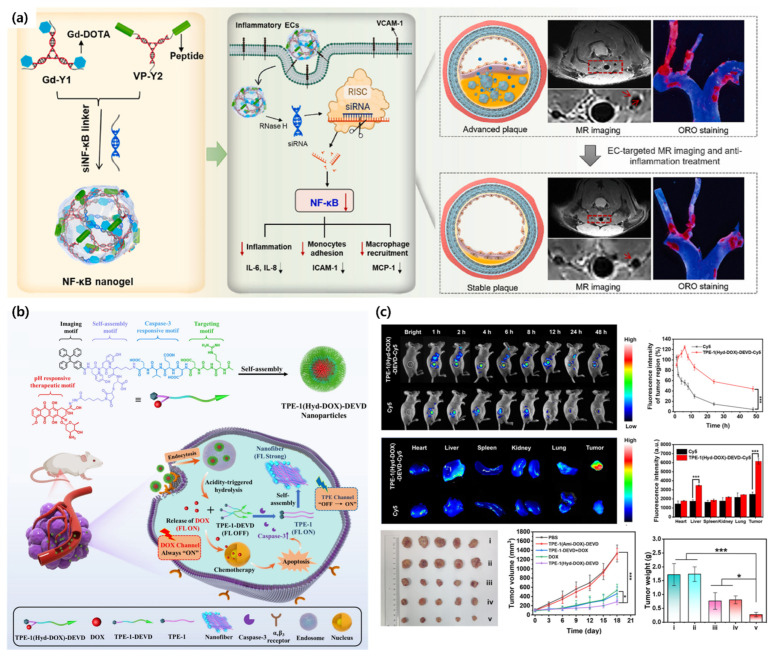
(**a**) Schematic of endothelium-targeted NF-κB nanogel enables MR imaging-guided, real-time monitoring and effective anti-inflammatory treatment of atherosclerosis by alleviation inflammation and regressing plaques [89]. (**b**) Schematic illustration of the self-reporting ratiometric AIEgen-peptide nanoprobe for selective activation of DOX and real-time monitoring of therapeutic efficacy in cancer targeting. (**c**) Fluorescence images of an in vivo tumor model demonstrating tumor targeting efficiency, biodistribution, and anti-tumor effects of TPE-1(Hyd-DOX)-DEVD NPs (One-way ANOVA, * *p* < 0.05 and *** *p* < 0.001) [94]. Copyright © 2025, American Chemical Society.

**Figure 14 biomimetics-10-00442-f014:**
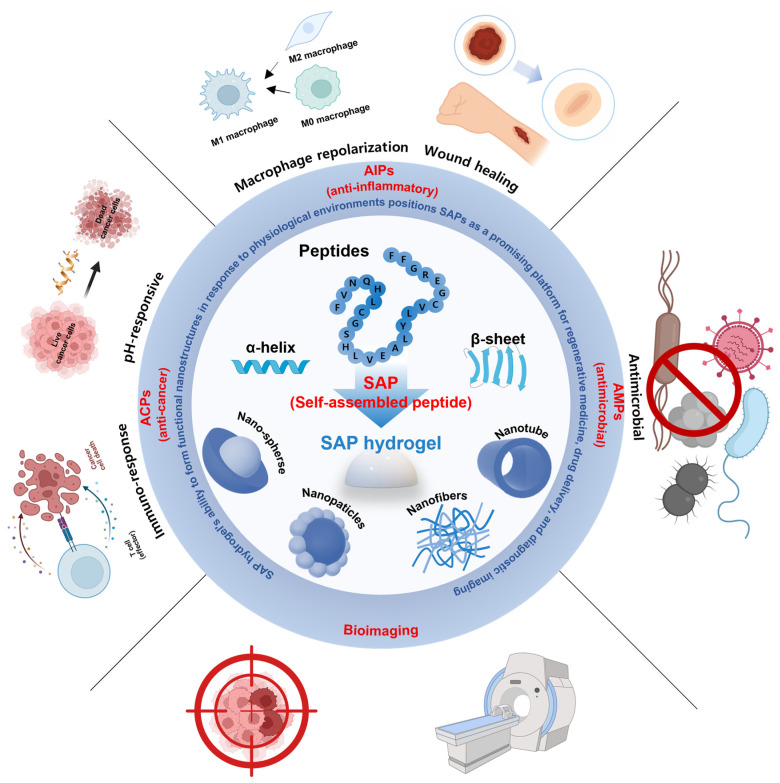
Graphical illustration of anti-inflammatory, antimicrobial, anticancer, and bioimaging.

**Table 1 biomimetics-10-00442-t001:** Summary of self-assembling peptide properties (anti-inflammatory).

Categories	Name (Sequence)	Bioactive Properties	Limitations	Ref.
Anti- Inflammatory	Biphenyl group peptide (Bip-KFRKAFKRFF)	- M2 to M1 macrophage polarization - Multivalent CD206 engagement	Long-term in vivo efficacy and safety remain unvalidated beyond in vitro and short-term animal model studies	[38]
	RADA16, nerve-promoted peptide (PPFLMLLKGSTR (PP))	- NSC proliferation enhancement - Neuronal differentiation promotion - Spinal cord matrix reconstruction	Translation to human clinical use remains unproven despite preclinical efficacy	[39]
	BiPM@IOK (Bismuthene nanosheet polyethyleneimine@IVNFOFLSK)	- pH-responsive drug release - Phototherapy-assisted fibroblast clearance - Synergistic immunomodulation in RA	- Requires laser irradiation for full therapeutic efficacy - Limiting practicality for non-invasive or systemic treatment approaches	[40]
	FFc5FF peptide (FFc5FF)	- Therapeutic NO generation and encapsulation - Anti-inflammatory gas signaling - Targeted biomedical gas delivery	- Requires alkaline pH and SNAP donor for NO generation - Limited endogenous activation under physiological conditions	[41]
	KLD1R peptide (Ac-KLDLKLDLKLDLR-CONH_2_)	- M2 macrophage polarization mediated by IL-10 - Alveolar bone regeneration - Diabetic tissue-specific immunomodulation	- Therapeutic efficacy depends on IL-10 loading - Hydrogel alone shows limited regenerative effect without cytokine	[42]
	Termed FP peptide (Nap-DFDFDEGPIRRSDS)	- TNF-α and ROS scavenging - Synovial inflammation suppression - Cartilage protection	- Relies on local intra-articular injection - Limited systemic applicability without invasive delivery	[43]
	BFD peptide (BPFFVLK-DSGLDSM)	- In situ nanofiber transformation - NF-κB pathway signaling inhibition - Targeted anti-inflammatory delivery	- Requires prolonged intracellular incubation to form functional nanofibers - Clinical translation and long-term effects remain unvalidated	[44]
	Hydrogel containing polypeptide (Ac2-26 (Ac))	- M2 macrophage polarization - Tissue regeneration in diabetic skin injuries - Angiogenesis stimulation	- Poor oral absorption and limited bioavailability - Low membrane permeability and rapid systemic clearance - Uncertainty in dose-response optimization for clincial application	[45]

**Table 2 biomimetics-10-00442-t002:** Summary of self-assembling peptide properties (antimicrobial).

Categories	Name (Sequence)	Bioactive Properties	Limitations	Ref.
Antimicrobial	Peptide nanofiber peptide (PNF) (KIIIIKYWYAF)	- Self-healing antimicrobial hydrogel formation - Broad-spectrum infection prevention with tissue compatibility	- Need for precise AgNP control - Risk of nanoparticle-induced toxicity	[54]
	Nanohydroxyapatite-loaded antimicrobial tripeptides (Fmoc-FRF, Dpha-FRF)	- Simultaneous bone regeneration and osteogenic differentiation - Broad-spectrum antimicrobial action	- Moderate mechanical strength - Limited long-term stability	[55]
	Q11 (AcQQKFQFQFEQQ-Am) + EH motif (n-GSEEEDHDHGEEDHHHE)	- Multivalent vaccine adjuvanticity - Immune activation	- Prolonged polymerization time - Delayed structural uniformity	[56]
	KR12 peptide (KRIVQRIKDFLR)	- Titanium based surface-integrated antimicrobial activity - Implant associated infection prevention	- Cytotoxicity at high concentrations - Reduced cell viability	[57]
	Jelleine-1 peptides (PFKLSLHL-NH_2_)	- Antibiotic-free antibacterial effect - Wound healing promotion	- Concentration-dependent gelation - Limited functionality under acidic conditions	[58]
	Jelleine-1 peptides (FFIHIKS)	- Effective against MRSA without the use of traditional antibiotics - Promotes rapid wound closure - Capable of forming stable nanofibrous hydrogels	- Short in vivo retention time - Environmental condition dependency - Long-term can cause immunogenic risks	[59]
	KLVFF self-healing nanofibrillar peptide (KLVFF)	- Enhanced antimicrobial efficacy through synergistic ion release - Hydrogel network reinforcement	- High dose requirement - Reduced in vivo efficiency - Limited therapeutic practicality	[60]
	C12G2 peptide (KKFFWDIL)	- Effective elimination of multi-resistant bacteria - Downregulation of pro-inflammatory cytokines - Acceleration of infected skin abscess healing in vivo	- Higher minimum inhibitory concentrations - High aggregation threshold requirement for hydrogel formation - Formulation complexity to optimize	[61]
	M(Myr)- 3FT F(Fmoc)- 3FT N(Nap)- 3FT (FFF+Tat peptide)	- Cell permeable membrane disruption - Intracellular multidrug- resistant bacteria eradication via ROS induction	- Nonspecific cellular uptake - Potential off-target effects	[62]

**Table 3 biomimetics-10-00442-t003:** Summary of bioimaging peptide properties (anticancer).

Categories	Name (Sequence)	Bioactive Properties	Limitations	Ref.
Anti-cancer	St-PSAPs (ST: Ac-AAAFFHH-NH_2_)	- pH-responsiveness to overcome tumor heterogeneity - Simple chemical modification	- Focus on in vitro model - Unclear drug release mechanism - Lack of long-term stability and immune response	[71]
	Pep-CS-LND hydrogel (Nap-GFFYK-CS-K(LND) KLAK)	- Redox responsiveness & mitochondrial targeting - Improved drug solubility	- Weak mechanical strength - Complexity of targeting mechanism - Limited control over drug release	[72]
	aP/IR@FMKB (Fmoc-KPLGLAGCRGDK)	- Target cancer cells to MMP-2 enzyme - Multimodal therapy integration - Prolonged local retention	- Limited mechanical strength - Challenges in drug release control - Restricted model testing - Complex fabrication process	[73]
	RADA16-R8 (Ac-(RADA)_4_-GG-RRRRRRRR–CONH_2_)	- Efficacy against multidrug-resistance cancer -Controlled release and biodegradability -Hemostatic properties	- Limited drug loading - Short-term stability - Complex fabrication	[74]
	Targeting GLUT1 glycopeptide (Naphthalene-FFKLVRRVR-glycosylation)-	- Effective cellular uptake and cancer inhibition - Lysosomal dysfunction induction - Broad applicability across cancer cell line	- Limited cell line testing - Lack of long-term safety data - Complex peptide synthesis - Dependence on specific mechanism	[75]
	RLDI & RQDT (RLDI: Ac-(RLDIKVEFCC)-CONH_2_ RQDT: Ac-(RQDTKTEYCC)-CONH_2_)	- Nanofiber scaffold design - Enhance DC functionality	- Complex peptide synthesis - Unclear optimal delivery method - Limited resolution of immunosuppressive environment	[76]
	N-Pep-Mn gel (N-Pep: Nap-FFYSV)	- Sustained drug release and local retention - Robust immune activation - Long-term immune memory effect	- Limited cancer model - Complex manufacturing process - Limitations of local administration - Partial resolution of immunosuppressive mechanism	[77]
	TEP-FFG-CRApY ((TPA-Eth-Py)-FFG–CRAWYQNpCALRR)	- Immunotherapy for low-immunogenic tumors - Light-controlled therapy	- Complex synthesis - Light dependency - Tumor microenvironment dependency	[78]

**Table 4 biomimetics-10-00442-t004:** Summary of bioimaging peptide properties (bioimaging).

Categories	Name (Sequence)	Bioactive Properties	Limitations	Ref.
Bioimaging	Polypeptide-TADF (Polypeptide: L-lysine repeated)	- Deep tissue penetration - Long fluorescence lifetime	- Single fluorophore reliance - Toxicity at high concentrations	[88]
	VCAM-1-targeted peptide (VHSPNK)	- Target endothelium to atherosclerosis diagnosis and treatment - Real-time treatment monitoring - High-resolution MR imaging	- Limited use of fluorescence imaging - Short monitoring duration - Lack of long-term stability	[89]
	^D^NPY(14)-ENTK-IR1048 (^D^NPY(14)-ENTK(Ac-CK[KDDDDKYD]HYNNPIWRQRY))	- To improve Blood-brain-barrier (BBB) permeability & long-term tumor tracking - Mitochondria-specific self-assembly	- Complex synthesis - Fluorophore dependency	[90]
	iRGD-DD, iRGD-WW ((CRGDKGPDC)-DD or WW)	- Target RGD to cancer imaging and therapy - High endosomal escape	- Reduced quantum yield - Lack of long-term stability - No comparison with alternative platform	[91]
	PNS/PEG-Ag_2_SQDs nanohybrids (PNS: Fmoc-FKKGSH)	- Strong NIR-II fluorescence - High photothermal efficiency - Tumor-specific accumulation	- Short-term fluorescence monitoring - Reliance on single imaging modality - Limited fluorescence persistence - Lack of long-term toxicity data	[92]
	CH08 depsipeptide (ValVValV)	- Nanovesicle structure - Multifunctionality and programmability	- Restricted to in vitro studies - Light-dependent limitation - Complexity of synthesis and scalability	[93]
	AIEgen-peptide nanoprobe (TPE-1(Hyd-DOX)-DEVD)	- Tumor-specific chemotherapy - Real time therapeutic monitoring - Ratiometric fluorescence	- FRET dependency - Lack of long-term toxicity data - Single fluorescence modality	[94]

## Data Availability

No new data were created or analyzed in this study.

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
