# Peer review of "Enhancing the Biological Functionality of Hydrogels Using Self-Assembling Peptides"

_biomimetics, 2025, doi:10.3390/biomimetics10070442_

Round 1
Reviewer 1 Report
Comments and Suggestions for Authors
The manuscript "Enhancing the Biological Functionality of Hydrogels Using Self-Assembling Peptides" presents an insightful summary of current advancements in peptide-based hydrogels. Manuscript is well-structured and effectively summarizes recent advancements in the corresponding field. The authors have provided comprehensive insights supported by clear illustrations. However, minor revisions are recommended to improve readability and scientific rigor:
- Although Section 2 adequately introduces diverse biofunctional properties of SAP-based hydrogels, the authors should enhance the depth by briefly discussing specific advantages and limitations of each method to provide a balanced perspective.
- Figures are valuable but require clearer and more detailed captions, explicitly describing their scientific significance and relevance to the corresponding text. Improving figure quality will significantly enhance readability. Importantly, for figures taken or adapted from other studies, proper citation within the figure legends must be clearly indicated.
- Numerous instances of incorrect hyphenation (e.g., poly-mer-based, nanotubes—closely mim-icking, matrix metalloprotein-ase-cleavable sequences) were observed throughout the manuscript. Authors are advised to carefully review and correct these errors for improved readability.
- The conclusion section would benefit significantly from a more comprehensive summary of key findings, explicit acknowledgment of existing limitations, and a deeper exploration of the authors' insights or recommendations for future research.
- Additionally, Section 2 is incorrectly labeled as "Materials & Method". This section title should be revised to accurately reflect its content.
Author Response
Reviewer #1 The manuscript "Enhancing the Biological Functionality of Hydrogels Using Self-Assembling Peptides" presents an insightful summary of current advancements in peptide-based hydrogels. Manuscript is well-structured and effectively summarizes recent advancements in the corresponding field. The authors have provided comprehensive insights supported by clear illustrations. However, minor revisions are recommended to improve readability and scientific rigor:
|
Comments 1: Although Section 2 adequately introduces diverse biofunctional properties of SAP-based hydrogels, the authors should enhance the depth by briefly discussing specific advantages and limitations of each method to provide a balanced perspective. |
Response 1: We appreciate the comment of the reviewer and revised the manuscript with adding limitation column and reorganized the tables by functional category to provide a more balanced and in-depth comparison of the functionality and limitation of each approach.
Table 2. Summary of self-assembled peptides properties (Antimicrobial). (Page 17) Table 3. Summary of bioimaging peptides properties (Anti-Cancer). (Page 24) Table 4. Summary of bioimaging peptides properties (Bioimaging). (Page 31)
|
Comments 2: Figures are valuable but require clearer and more detailed captions, explicitly describing their scientific significance and relevance to the corresponding text. Improving figure quality will significantly enhance readability. Importantly, for figures taken or adapted from other studies, proper citation within the figure legends must be clearly indicated. |
Response 2: Thank you for your helpful feedback regarding the figures. In response, we have revised the figure captions to provide clearer and more detailed descriptions, highlighting their scientific relevance to the main text. We have also improved the figure quality to enhance overall readability. Additionally, for figures that were adapted from previous studies, we have included appropriate citations directly within the figure legends, as required.
Figure 4. (a) Schematic representation of the systematic treatment strategy using BiPM@IOK hydrogel, which combines anti-inflammatory activity with targeted delivery for enhanced rheumatoid arthritis (RA) therapy. (b) Quantification of TNF-α and IL-1β of synovium tissue and Safranin O staining of joints section after treatment with BiPM@IOK hydrogel [40]. (c) Schematic representation of the multifunctional hydrogel containing polypeptide with an-ti-inflammatory and wound healing function. (d) Representative images of the wound healing progression over the treatment and relative gene expression of TNF-α and IL-1β [45].
Figure 6. Schematic the synthesis of an antimicrobial hydrogel composed of quaternized chitosan (QCS), aldolized hyaluronic acid (AHA), peptide nanofibers (PNFs), and silver nanoparticles (AgNPs) (a), with its antibacterial properties evaluated through the effects of varying PNF/AgNPs content on E.coli and S. aureus, along with corresponding statistical analyses (b) [54]. Copyright © 2024, John Wiley and Sons.
Figure 7. (a) Schematic illustration of PGK-CuTN coating on Titanium substrate using multifunctional SAPs contain with antimicrobial, angiogenesis, anti-inflammatory, and osteogenesis. (b) Representative images of TN and PGK-CuTN after 12 hours of bacterial colony rolling cultivation, and histological analysis of bone defect implants via H&E and Masson's trichrome staining [57]. Copyright © 2025, Elsevier. (c) Design and functional evaluation of self-assembled nonapeptide (F3FT and N3FT) with dual capabilities of cell penetration and antibacterial activity. (d) Representative FE-SEM and TEM images showing the membrane-disruptive effects of F3FT and N3FT on S. aureus [62].
Figure 9. (a) Schematic illustration and functional evaluation size-transformable peptide self-assembling particles (St-PSAPs) designed for tumor microenvironment (TME) pH-responsive, tumor-specific therapy. (b) Cellular uptake of St-PSAPs was assessed under pH 6.4 and pH 7.4 conditions across various cell types, demonstrating enhanced uptake in acidic TME [71]. Copy-right © 2025, Elsevier.
Figure 10. (a) Schematic illustration of the preparation of octa-arginine (R8)-modified RADA16 (RR) SAPs nanofiber hydrogel and mechanisms and therapeutic benefits as a doxorubicin (DOX) delivery system for anti-cancer. (b) Representative images of dissected tumors, quantification of tumor weights, and tumor growth curves to evaluate the in vivo anti-cancer effect of SAPs. [74]. (c) Proposed model illustrating SAPs as a novel vaccine delivery platform that forms nanostructures to load various vaccine types, support dendritic cell (DC) activation, enable sustained antigen release, and enhance CD4+/CD8+ T cell responded for effective anti-tumor immunity. (d) Confocal microscopy imaging showing DC recruitment stimulated by RLDI- or RQDT-OVA treatment, along with representative images of dissected tumor, quantification of tumor weight and volume demonstrating the therapeutic efficacy of peptide hydrogel-loaded DC vaccines [76].
Figure 12. (a) Schematics illustration of the characterization and conformation-dependent photophysical properties of α-helical polypeptide nanoprobes. (b) Enhanced imaging contrast and deep tissue penetration achieved by α-helical polypeptide nanoprobes [88]. Copyright © 2025, American Chemical Society.
Figure 13. (a) Schematic of endothelium-targeted NF-κB nanogel enables MR imaging-guided, real-time monitoring and effective anti-inflammatory treatment of atherosclerosis by alleviation inflammation and regressing plaques [89]. (b) Schematic illustration of the self-reporting ratiometric AIEgen-peptide nanoprobe for selective activation of DOX and re-al-time monitoring of therapeutic efficacy cancer targeted. (c) Fluorescence images of an in vivo tumor model and demonstrating tumor targeting efficiency, biodistribution, and anti-tumor effects of TPE-1(Hyd-DOX)-DEVD NPs [94]. Copyright © 2025, American Chemical Society.
|
Comments 3: Numerous instances of incorrect hyphenation (e.g., poly-mer-based, nanotubes—closely mim-icking, matrix metalloprotein-ase-cleavable sequences) were observed throughout the manuscript. Authors are advised to carefully review and correct these errors for improved readability. |
Response 3: We appreciate the feedback given by the reviewer and performed a complete check of the whole manuscript and amended such errors throughout.
|
Comments 4: The conclusion section would benefit significantly from a more comprehensive summary of key findings, explicit acknowledgment of existing limitations, and a deeper exploration of the authors' insights or recommendations for future research. |
Response 4: We are grateful for the reviewer’s encouraging comments. In response to the suggestion about future directions, we have included a new section titled “4. Challenges and Future Perspectives” to elaborate on existing limitations and explore opportunities for further research, improving the overall depth of the manuscript.
|
Comments 5: Additionally, Section 2 is incorrectly labeled as "Materials & Method". This section title should be revised to accurately reflect its content. |
Response 5: We appreciate the review’s helpful comments. We have revised the title of Section 2 form “Materials & Method” to “Bioactive Properties of Self-Assembling Peptides” to more accurately reflect the content (Line 110). |

Reviewer 2 Report
Comments and Suggestions for Authors
This review manuscript provides a systematic overview of strategies for enhancing the biological hydrogels composed of self-assembling peptides (SAPs), covering the applications in the field of anti-inflammatory, antimicrobial activity, anticancer therapy, and bioimaging. Each section is well-organized by topic, with clear explanations linking peptide sequences, self-assembly, and biological outcomes to enhance clarity. Although the discussion of future directions is limited, this manuscript is well-written and provides a broad contribution to this field. So it is recommended for publication.
Author Response
Reviewer #2 Comments : This review manuscript provides a systematic overview of strategies for enhancing the biological hydrogels composed of self-assembling peptides (SAPs), covering the applications in the field of anti-inflammatory, antimicrobial activity, anticancer therapy, and bioimaging. Each section is well organized by topic, with clear explanations linking peptide sequences, self-assembly, and biological outcomes to enhance clarity. Although the discussion of future directions is limited, this manuscript is well-written and provides a broad contribution to this field. So it is recommended for publication |
Response : We appreciate the reviewer’s positive feedback. To address the comment on future directions, we added a new section, “4. Challenges and Future Perspectives” to discuss current limitations and future research opportunities, enhancing the depth of the manuscript. |

Reviewer 3 Report
Comments and Suggestions for Authors
The manuscript provides a comprehensive review of self-assembling peptides (SAPs) and their applications in enhancing the biological functionality of hydrogels. The topic is timely and relevant, given the growing interest in biomimetic materials for biomedical applications. The review is well-structured, covering anti-inflammatory, antimicrobial, anti-cancer, and bioimaging functionalities of SAP-based hydrogels. However, there are areas where the manuscript could be improved to enhance clarity, depth, and impact.
- Some mechanisms (e.g., membrane disruption by antimicrobial peptides) are repeated across sections, which could be streamlined. Consolidate overlapping mechanistic discussions and refer back to earlier sections where applicable.
- The anti-inflammatory and antimicrobial sections are detailed, but the bioimaging section feels comparatively brief. Expand the bioimaging section with more examples of clinical applications or emerging technologies (e.g., AIEgens in diagnostics).
- Terms like "multifunctionality" and "stimuli-responsive" are used frequently but could be better defined early in the manuscript. Include a glossary or introductory paragraph clarifying key terms.
- Some figures (e.g., Figure 1) are overly simplistic and could benefit from more detailed annotations or legends. Redesign figures to highlight specific peptide sequences, assembly processes, or functional outcomes.
- The manuscript does not compare SAP-based hydrogels with other advanced biomaterials (e.g., nanocomposites, synthetic polymers). Add a table or paragraph comparing SAPs to alternative strategies in terms of efficacy, cost, and clinical feasibility.
- Some sentences are overly long or awkwardly phrased (e.g., "These peptides attenuate inflammatory responses through three primary biological mechanisms: receptor-mediated signaling suppression, oxidative stress mitigation, and protease-resistant structural stabilization").
- Ensure all cited studies are up-to-date (e.g., include 2024–2025 publications where relevant).
- While the review summarizes many studies, it lacks critical evaluation. For example, the limitations of SAPs (e.g., scalability, cost, stability in vivo) are not thoroughly discussed. Please add a dedicated section or subsection addressing challenges and future directions, including translational barriers and potential solutions.
Author Response
Reviewer #3 The manuscript provides a comprehensive review of self-assembling peptides (SAPs) and their applications in enhancing the biological functionality of hydrogels. The topic is timely and relevant, given the growing interest in biomimetic materials for biomedical applications. The review is well-structured, covering anti-inflammatory, antimicrobial, anti-cancer, and bioimaging functionalities of SAP-based hydrogels. However, there are areas where the manuscript could be improved to enhance clarity, depth, and impact.
|
Comments 1: Some mechanisms (e.g., membrane disruption by antimicrobial peptides) are repeated across sections, which could be streamlined. Consolidate overlapping mechanistic discussions and refer back to earlier sections where applicable. |
Response 1: We appreciate the reviewer’s valuable suggestion. In response, we carefully reviewed the manuscript and reduced redundant descriptions of overlapping mechanisms in the relevant sections. Where appropriate, we streamlined the mechanistic discussions and referred back to earlier explanations to avoid unnecessary repetition. We sincerely thank the reviewer for the insightful feedback, which has helped improve the overall clarity and quality of the manuscript.
|
Comments 2: The anti-inflammatory and antimicrobial sections are detailed, but the bioimaging section feels comparatively brief. Expand the bioimaging section with more examples of clinical applications or emerging technologies (e.g., AIEgens in diagnostics). |
Response 3:
2.4.2. Peptides for Enhancing Bioimaging Properties in Hydrogels Conventional molecular imaging tools are often limited by short penetration depth, to improve deep tissue penetration, Chu, Yang, et al. developed rigid α-helical polypetide nanoprobes (L-PTN) with thermally activated delayed fluorpolypeptides scaffolds, providing orange emission (> 600 nm) and micro-second-scale fluorescence lifetimes... [88] Building on the need for both diagnostic precision and therapeutic intervention, Guo, Yuanyuan, et al. presented the development of an endothelium-targeted NF-κB siRNA nanogel (VP-Gd-NF-NG) for atherosclerosis diagnosis and treatment....[89] While vascular targeting has shown promise, brain-target imaging presents unique challenges due to blood-brain barrier (BBB).... [90] Complementing fluorescence-based imaging, photoacoustic imaging has gained at-tention for its ability to provide deeper penetration and spatial resolution...[91] Expanding the utility of NIR-II imaging further, Luan, Xin, et al. presented a 2D na-noplatform, peptide nanosheets (PNS)/PEG-Ag2S quantum dots (QDs) nanohybrids, de-signed for near-infrared-II (NIR-II) fluorescent bioimaging and photothermal therapy (PTT) of tumors...[92] The rationale for developing bioimaging technology by combining Aggrega-tion-Induced Emission luminogens (AIEgens) and peptides is to combine the excellent fluorescent properties of AIEgens with the biocompatibility and functionality of peptides to enable high-sensitivity, high-definition, and target-directed imaging...[93] Taking the concept further into therapeutic monitoring, Pei, Shicheng, et al. underscored the potential of AIEgen-peptide nanoprobes to enhance cancer treatment precision by in-tegrating bioimaging and chemotherapy...[94]
|
Comments 3: Terms like "multifunctionality" and "stimuli-responsive" are used frequently but could be better defined early in the manuscript. Include a glossary or introductory paragraph clarifying key terms. |
Response 2: We appreciate the reviewer’s valuable suggestion. To improve clarity, we have included an introductory explanation early in “Introduction” section defining key terms such as “multifunctionality” and “stimuli-responsive.”
In the context of peptide-based biomaterials, multifunctionality refers to the capacity of a single self-assembling peptide (SAP) system to perform multiple biological and physico-chemical functions simultaneously or sequentially.... [Line 69~]
Multifunctionality is typically achieved through the modular design of peptide sequences..... [Line 79~]
|
Comments 4: Some figures (e.g., Figure 1) are overly simplistic and could benefit from more detailed annotations or legends. Redesign figures to highlight specific peptide sequences, assembly processes, or functional outcomes. |
Response 4: We sincerely thank the reviewer for the constructive feedback regarding the figures. In particular, we agree that “Figure 1” could be improved to better illustrate the key concepts.
|
Comments 5: The manuscript does not compare SAP-based hydrogels with other advanced biomaterials (e.g., nanocomposites, synthetic polymers). Add a table or paragraph comparing SAPs to alternative strategies in terms of efficacy, cost, and clinical feasibility. |
Response 5:
Table 2. Summary of self-assembled peptides properties (Antimicrobial). (Page 17) Table 3. Summary of bioimaging peptides properties (Anti-Cancer). (Page 24) Table 4. Summary of bioimaging peptides properties (Bioimaging). (Page 31) 4. Challenges and Future Perspectives (Page 33-34)
|
Comments 6: Some sentences are overly long or awkwardly phrased (e.g., "These peptides attenuate inflammatory responses through three primary biological mechanisms: receptor-mediated signaling suppression, oxidative stress mitigation, and protease-resistant structural stabilization"). |
Response 6: We thank the reviewer for the helpful comment. We revised the manuscript to improve clarity by shortening and restructuring complex sentences.
In this context, anti-inflammatory peptides (AIPs) constitute a promising class of biomaterials that leverage supramolecular chemistry and immunomodulatory functions to deliver targeted and sustained therapeutic effects. They modulate inflammation through various mechanisms, such as suppression of receptor-mediated signaling, mitigation of oxidative stress, and stabilization against protease degradation. [32,33]. [Line 132-136]
|
Comments 7: Ensure all cited studies are up-to-date (e.g., include 2024–2025 publications where relevant). |
Response 7: We acknowledge the suggestion made by the reviewer and revised up-to-date research articles. 2.1.2. Peptides for Enhancing Anti-Inflammatory Properties in Hydrogel Moreover, Sun et al. reported peptide-based composite hydrogels (CRP) for spinal cord injury (SCI) repair...[39] As another example of inflammation-modulatory properties, Liu et al. addressed trans-formable peptide nanoplatform (BP-FFVLK-DSGLDSM, BFD) designed to treat rheumatoid arthritis (RA) by targeting the NF-κB/IκBα signaling pathway...[44] Addressing diabetic wound repair, Lu et al. utilized a hyaluronic acid (HA)-based hydrogel loaded with the Ac2-26 (Ac) peptide, which exhibited good mechanical properties, self-healing ability, and strong adhesion...[45]
2.2.2. Peptides for Enhancing Antimicrobial Properties in Hydrogels Based on these mechanisms, Zhang et al. introduced an injectable, self-healing antimicrobial hydrogel composed of quaternized chitosan (QCS) and aldolized hyaluronic acid (AHA), integrated with self-assembling peptide nanofibers (PNFs) and ultrasmall silver nanoparticles (AgNPs)…[54] Meanwhile, Eliza et al. reported a self-assembling peptide-based vaccine antigen, Q11-EH, targeting Gram-positive ESKAPE pathogens, including vancomycin-resistant Enterococcus faecium and methicillin-resistant Staphylococcus aureus....[56] Moreover, Xu et al. discussed multifunctional PGK-CuTN coatings on titanium substrates for the treatment of infected bone defects....[57] Furthermore, Zhu et al. reported self-assembled nanopeptides, F3FT and N3FT, designed to combat intracellular bacterial infections and antimicrobial resistance...[62]
2.3.2. Peptides for Enhancing Anti-Cancer Properties in Hydrogels Tumor microenvironment (TME) exhibits distinct physiological and biochemical characteristics compare healthy tissues, such as low pH, elevated levels of reactive oxygen species (ROS), and increased activity of specific enzymes such as matrix metalloproteinase (MMPs) and other proteolytic enzymes...[71] To improve intratumoral penetration and retention, Wang, Ying, et al. developed an octa-arginine (R8)-modified RADA16 (RR) self-assembling peptide nanofiber hydrogel, demonstrating its capabilities in selective tumor penetration, sustained drug release, and hemostasis....[74] Expanding on intracellular targeting, Wu, Xia, et al. demonstrated that the glycopeptide selectively degraded GLUT1 (glucose transporter 1), which is overexpressed in most cancer cells, thereby exerting a tumor-suppressive effect in HCT-116 cells...[75] Recognizing the limitation posed by poorly immunogenic tumor, Wang, He, et al. designed TEP-FFG-CRApY effectively penetrates prostate cancer (PCa) and induces ferroptosis, thereby inhibiting tumor growth...[78]
2.4.2. Peptides for Enhancing Bioimaging Properties in Hydrogels Conventional molecular imaging tools are often limited by short penetration depth, to improve deep tissue penetration, Chu, Yang, et al. developed rigid α-helical polypetide nanoprobes (L-PTN) with thermally activated delayed fluorescence (TADF) for time-resolved, high-contrast bioimaging... [88] Building on the need for both diagnostic precision and therapeutic intervention, Guo, Yuanyuan, et al. presented the development of an endothelium-targeted NF-κB siRNA nanogel (VP-Gd-NF-NG) for atherosclerosis diagnosis and treatment....[89] While vascular targeting has shown promise, brain-target imaging presents unique challenges due to blood-brain barrier (BBB).... [90] Complementing fluorescence-based imaging, photoacoustic imaging has gained attention for its ability to provide deeper penetration and spatial resolution...[91] Expanding the utility of NIR-II imaging further, Luan, Xin, et al. presented a 2D nanoplatform, peptide nanosheets (PNS)/PEG-Ag2S quantum dots (QDs) nanohybrids, designed for near-infrared-II (NIR-II) fluorescent bioimaging and photothermal therapy (PTT) of tumors...[92] The rationale for developing bioimaging technology by combining Aggregation-Induced Emission luminogens (AIEgens) and peptides is to combine the excellent fluorescent properties of AIEgens with the biocompatibility and functionality of peptides to enable high-sensitivity, high-definition, and target-directed imaging...[93] Taking the concept further into therapeutic monitoring, Pei, Shicheng, et al. underscored the potential of AIEgen-peptide nanoprobes to enhance cancer treatment precision by integrating bioimaging and chemotherapy...[94]
|
Comments 8: While the review summarizes many studies, it lacks critical evaluation. For example, the limitations of SAPs (e.g., scalability, cost, stability in vivo) are not thoroughly discussed. Please add a dedicated section or subsection addressing challenges and future directions, including translational barriers and potential solutions. |
Response 8: We thank the reviewer for the encouraging feedback and thoughtful suggestions. In response to the comment on the limited discussion of future directions, we have included a new section, “4. Challenges and Future Perspectives (Page 33-34)” to outline current limitations and highlight future research opportunities. We hope this addition improves the manuscript’s depth and overall quality. |

Round 2
Reviewer 3 Report
Comments and Suggestions for Authors
The manuscript has been revised well according to the previous comments.